# FlashBind: Towards Accurate and Efficient Structure-based Virtual Screening

## Abstract

Accurate prediction of protein-ligand interactions is central to computational drug discovery. Recent foundation models such as Boltz-2 have achieved remarkable accuracy in binding affinity prediction, yet their prohibitive computational cost remains a major barrier to large-scale virtual screening. Here we introduce **FlashBind**, a lightweight structure-based model that achieves a **50× speedup** over Boltz-2 at inference time by replacing expensive structure prediction with a fast docking model and substituting costly PairFormer modules with a streamlined EGNN architecture. FlashBind attains early enrichment competitive with Boltz-2 on standard virtual screening benchmarks and demonstrates strong generalization to enzyme-substrate specificity prediction. To evaluate real-world applicability, we apply FlashBind to target-based antibiotic screening against the essential bacterial proteins in *E. coli* and show that FlashBind substantially outperforms Boltz-2 and other virtual screening baselines. Notably, several top-ranked candidates exhibit potent inhibition of DnaG and effective bacterial growth inhibition against E. coli in wet-lab validation. Together, these results demonstrate that FlashBind bridges the gap between accuracy and efficiency, enabling ultra-fast and accurate screening of massive chemical libraries for drug discovery.

## 1 Introduction

The discovery of novel bioactive small molecules is a fundamental pursuit in pharmaceutical science, yet it remains hindered by the vast chemical space, which is estimated to contain more than $10^{60}$ drug-like compounds (Reymond, 2015). High-throughput virtual screening serves as the critical filter in this process, aiming to identify potential binders from massive chemical libraries before experimental validation. Currently, structure-based virtual screening methods are divided into two categories. Physics-based docking approaches, such as AutoDock Vina (Trott & Olson, 2009), GNINA (McNutt et al., 2021), and Glide (Halgren et al., 2004), are computationally expensive and struggle to scale to ultra-large chemical libraries. On the other hand, deep learning models offer greater throughput but often suffer from limited generalizability across diverse targets.

In recent years, this lack of generalizability has been reshaped by the emergence of foundation models trained on immense biological datasets. Models such as Boltz-2 (Passaro et al., 2025) have achieved remarkable accuracy in predicting protein-ligand complex structures and binding affinities. However, these gains come with prohibitive computational cost. The intricate architecture of such models, often relying on expensive recycling mechanisms and PairFormer modules, poses a major barrier to their deployment in large-scale virtual screening campaigns. For instance, processing a single protein-ligand complex with Boltz-2 requires approximately 35 seconds, a timeframe that renders the screening of billion-scale libraries computationally intractable. Therefore, a critical gap remains: the field lacks a solution that can match the predictive fidelity of foundation models while maintaining the throughput required for industrial-scale discovery.

To address this challenge, we introduce **FlashBind**, a lightweight geometric deep learning framework designed to bridge the gap between accuracy and efficiency. FlashBind achieves the accuracy-efficiency trade-off by structurally decoupling the screening process into two streamlined stages: rapid structure generation and geometric scoring. Instead of relying on computationally intensive diffusion-based generation or end-to-end

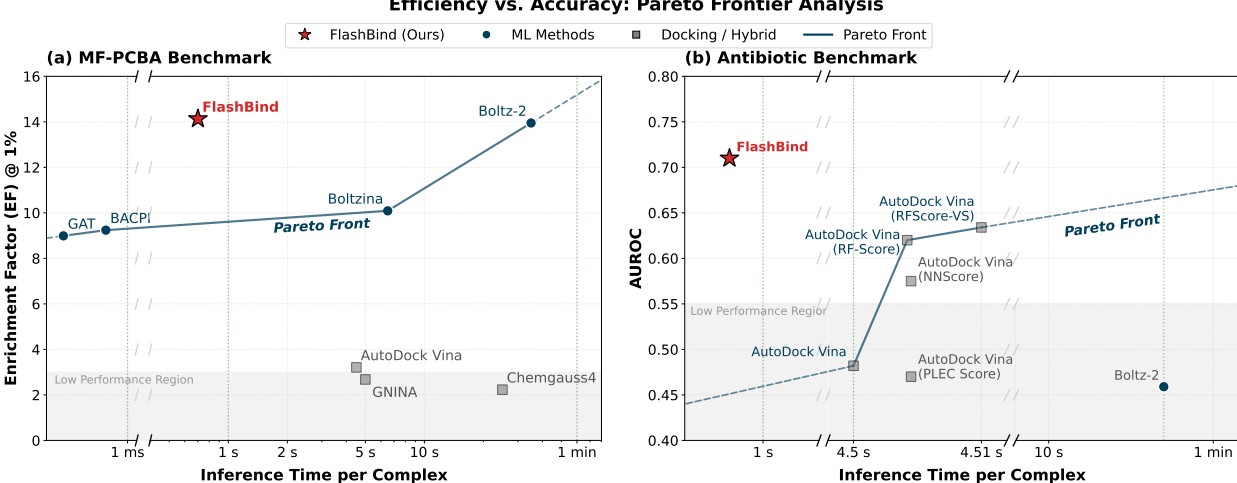

Figure 1: **Computational efficiency vs. screening accuracy.** FlashBind (star) occupies the optimal efficiency-accuracy trade-off, matching Boltz-2's top-1% early enrichment with 50× faster inference (0.7s vs. 35s). The inference time is measured on a single NVIDIA L40S GPU.

folding, our framework utilizes a fast docking model (FABind+) (Gao et al., 2025) to provide a physically plausible structural prior. This structure is then processed by a highly efficient $E(3)$-equivariant Graph Neural Network (EGNN) (Satorras et al., 2022), which replaces heavy attention mechanisms to capture local physical interactions within the binding pocket.

This simple design proves remarkably effective: across virtual screening, enzyme-substrate specificity, and a prospective antibiotic discovery campaign, FlashBind is competitive with or exceeds far heavier foundation models at a fraction of the computational cost. We summarize our main contributions as follows:

- **A lightweight, ultra-fast screening framework.** We replace expensive diffusion-based structure generation and PairFormer-based scoring with a fast docking model (FABind+) (Gao et al., 2025) followed by an $E(3)$-equivariant graph neural network (Satorras et al., 2022), yielding a **50-fold inference speedup** over Boltz-2 ($\sim 0.7\,\text{s}$ vs. $\sim 35\,\text{s}$ per complex on a single NVIDIA L40S GPU).

- **Competitive early enrichment at a fraction of the cost.** On the MF-PCBA benchmark (Buterez et al., 2023), FlashBind attains early enrichment competitive with the foundation model Boltz-2, matching it at the top 1% (EF@1% of 14.13 vs. 13.95), while substantially outperforming physics-based and sequence-based baselines. Ablation studies confirm that neither diffusion sampling nor a heavy PairFormer trunk is necessary to attain this accuracy.

- **Generalization to enzyme-substrate specificity.** Under the challenging "unknown enzyme & substrate" setting of the ESIBank benchmark (Cui et al., 2025), FlashBind reaches an AUROC of 0.7229, on par with the specialized EZSpecificity (0.7198) and well above the sequence-based ESP (Kroll et al., 2023), while surpassing Boltz-2 on data-scarce enzyme families.

- **Prospective experimental validation.** On a structure-based antibiotic benchmark against essential *E. coli* proteins (Wong et al., 2022), FlashBind attains an AUROC of 0.71 where Boltz-2 is near-random (0.46). In a prospective campaign against *E. coli* DNA primase (DnaG), we screened 9289 compounds, experimentally assayed 136, and confirmed 10 active inhibitors (a 7.4% hit rate), 4 of which further exhibited whole-cell antibacterial activity.

Together, these results establish FlashBind as a scalable and accurate framework for ultra-fast virtual screening of massive chemical libraries.

## 2 Related Work

**Structure-based Virtual Screening.** Virtual screening seeks to identify, from chemical libraries now reaching the billion-compound scale, the small subset of molecules likely to bind a given target. The classical workhorse is molecular docking (Trott & Olson, 2009; Halgren et al., 2004), which predicts a ligand's binding pose and energy within a pocket. Docking is, however, computationally intensive, making exhaustive screening of modern libraries impractical. Supervised learning offers a faster alternative: discriminative models trained to classify protein-ligand pairs as binding or non-binding trade physical interpretability for throughput, but often generalize poorly to unseen targets. A more recent paradigm reframes screening as dense retrieval (Radford et al., 2021): methods such as DrugCLIP (Gao et al., 2023) contrastively align separate protein and ligand encoders into a shared embedding space, enabling library embeddings to be precomputed and screening to reduce to fast similarity search. Geometric deep learning instead operates directly on 3D complex structures for higher fidelity, typically by representing the complex as an atomic graph and enforcing geometric symmetries (Satorras et al., 2022); this is the regime in which FlashBind operates, with the key distinction that we decouple a fast docking prior from a lightweight equivariant scorer to retain structural accuracy without per-compound conformational sampling.

**Enzyme-Substrate Specificity.** Predicting which substrates an enzyme acts upon is a functional task governed by catalytic alignment rather than thermodynamic stability alone, and it has historically been treated separately from binding prediction. Sequence-based approaches such as ESP (Kroll et al., 2023) pair enzyme representations with molecular fingerprints to classify enzyme-substrate pairs, but discard the 3D geometry of the active site. More recent structure-aware methods exploit predicted complex geometry: EZSpecificity (Cui et al., 2025) introduces a task-specific cross-attention architecture together with the ESIBank benchmark, establishing the current state of the art under stringent unknown-enzyme-and-substrate splits. Our work shows that a general-purpose equivariant encoder matches this specialized model, suggesting that the geometric features useful for binding transfer to catalytic specificity.

**Machine Learning for Antibiotic Discovery.** Machine learning has emerged as a powerful tool against the slowing pace of antibiotic discovery. Early efforts were predominantly ligand-based: deep classifiers trained on phenotypic growth-inhibition data prioritize compounds by predicted antibacterial activity directly from molecular structure (Stokes et al., 2020). Such phenotype-driven models are agnostic to mechanism and offer little insight into the molecular target. Structure-based formulations address this by scoring compounds against specific essential bacterial proteins; the benchmark of Wong et al. (Wong et al., 2022) couples docking against a panel of essential *E. coli* targets with experimental inhibition assays, providing a realistic and mechanistically grounded testbed. We adopt this structure-based setting and further close the loop with a fully prospective wet-lab campaign.

**Bridging the Accuracy-Speed Gap.** A central obstacle in structure-based screening is the trade-off between accuracy and throughput. Boltz-2 (Passaro et al., 2025) marked a turning point, reportedly approaching the accuracy of free-energy perturbation for protein-ligand affinity while being orders of magnitude cheaper, making high-accuracy prediction feasible in discovery settings. Even so, its per-complex inference cost remains a bottleneck for routine large-library screening, motivating efforts to accelerate the architecture. Derivative methods carry their own constraints: Boltzina (Furui & Ohue, 2025), for instance, requires predefined pocket information and retains the original Boltz trunk, leaving substantial headroom on speed. FlashBind takes a more aggressive stance, replacing both the diffusion-based structure generator and the heavy trunk with a docking oracle and an equivariant graph network, which removes the trunk bottleneck entirely while preserving early-enrichment accuracy.

## 3 Method

### 3.1 Problem Formulation

We address structure-based virtual screening: identifying the small subset of compounds in a large chemical library that bind a given protein target. This hit-discovery setting prioritizes ranking active compounds

(binders) above inactive decoys over precisely quantifying binding strength, and we therefore formulate it as a **binary classification** task. Formally, given a protein amino acid sequence $\mathcal{S}_p$ and a ligand SMILES string $\mathcal{S}_l$, FlashBind learns a mapping $\mathcal{F} : (\mathcal{S}_p, \mathcal{S}_l) \rightarrow p_{\text{bind}} \in [0, 1]$, where $p_{\text{bind}}$ is the predicted probability that the ligand binds the target. As described below, $\mathcal{F}$ is realized by mapping the raw inputs to a 3D complex $(\mathbf{X}, \mathbf{H})$ via fast docking, cropping it to the binding interface, encoding the result as a geometric graph $\mathcal{G}$, and scoring $\mathcal{G}$ with an $E(3)$-equivariant network. The same encoder is task-agnostic: replacing the classification head with a regression head yields continuous affinity prediction ($y_{\text{affinity}}$), which we treat as an auxiliary task and detail in Appendix A.

## 3.2 The FlashBind Architecture

FlashBind resolves the accuracy-efficiency trade-off by structurally decoupling screening into two streamlined stages: rapid structure generation followed by geometric scoring (Fig. 2b). This avoids both diffusion-based conformational sampling and end-to-end folding, the two dominant bottlenecks in foundation-model pipelines.

**Structure generation.** For inputs given only as a sequence and a SMILES string, we obtain a protein structure through a hierarchical retrieval cascade that prioritizes experimental fidelity: we first query the PDB (Berman et al., 2000) for an experimental structure matching the target at 100% identity, otherwise select the highest-pLDDT model from the AlphaFold Database (Varadi et al., 2021), and only as a last resort predict the structure de novo with Boltz-2x (Passaro et al., 2025). In practice, over 60% of targets are resolved directly from the PDB or AlphaFold, sharply reducing prediction overhead. The ligand is then docked into this structure with FABind+ (Gao et al., 2025), a regression-based docking model chosen for its speed ($< 0.7$ s per complex), producing a 3D complex $(\mathbf{X}, \mathbf{H})$ with atomic coordinates $\mathbf{X} \in \mathbb{R}^{N \times 3}$ and features $\mathbf{H} \in \mathbb{R}^{N \times d}$. Crucially, this step supplies a physically plausible structural prior at a fraction of the cost of diffusion-based generation.

**Graph construction.** To focus the encoder on the binding interface, an adaptive cropping function $\mathcal{F}_{\text{crop}}$ isolates the local pocket: residues are added greedily by proximity until an atom budget ($B_a = 2048$) or residue budget ($B_r = 512$) is reached, after which residues beyond 20Å are removed. The cropped complex is converted into a multi-relational geometric graph $\mathcal{G} = (\mathcal{V}, \mathcal{E}, \mathbf{h}, \mathbf{x})$, whose nodes $\mathcal{V}$ are protein and ligand atoms. Node features $\mathbf{h}$ concatenate pre-trained ESM-3 (Hayes et al., 2025) protein embeddings with TorchDrug (Zhu et al., 2022) ligand descriptors computed via RDKit (Landrum et al., 2025). Following MEAN (Kong et al., 2023), the edge set $\mathcal{E}$ captures interactions at multiple scales: **internal edges** for intra-molecular covalent topology, **external edges** for non-covalent protein-ligand contacts ($< 10$Å), and **auxiliary edges** that inject global structural context through global nodes. Full node featurization, the budget-constrained cropping algorithm, and the complete edge taxonomy with its priority scheme are detailed in Appendix B.1, B.2, and B.3, respectively.

**Equivariant scoring network.** The graph $\mathcal{G}$ is processed by an $E(3)$-equivariant graph neural network (EGNN) (Satorras et al., 2022) of $L = 5$ layers with hidden dimension 192. Enforcing $E(3)$-equivariance makes the prediction invariant to the arbitrary rotation and translation of the docked pose, removing any need for orientation augmentation. The network produces a pooled graph representation $z_{\mathcal{G}}$, which a task-specific MLP head maps to the binding probability $p_{\text{bind}}$ (and, for the auxiliary regression task, to $y_{\text{affinity}}$).

## 3.3 Data Curation

A reliable binding signal can only be learned from high-quality supervision, yet high-throughput screening (HTS) data are notoriously noisy. We therefore curate the training data with a multi-stage filtration pipeline (Fig. 2a).

**Virtual screening dataset.** We build our screening corpus from PubChem BioAssays (Kim et al., 2022), explicitly counteracting the high false-positive rates endemic to HTS. The pipeline begins with an *assay-level* filter that retains only confirmatory and primary screens containing more than 100 compounds with hit rates below 10%; assays targeting the same protein (matched by UniProt ID (Consortium, 2024)) are merged to

**(a)**

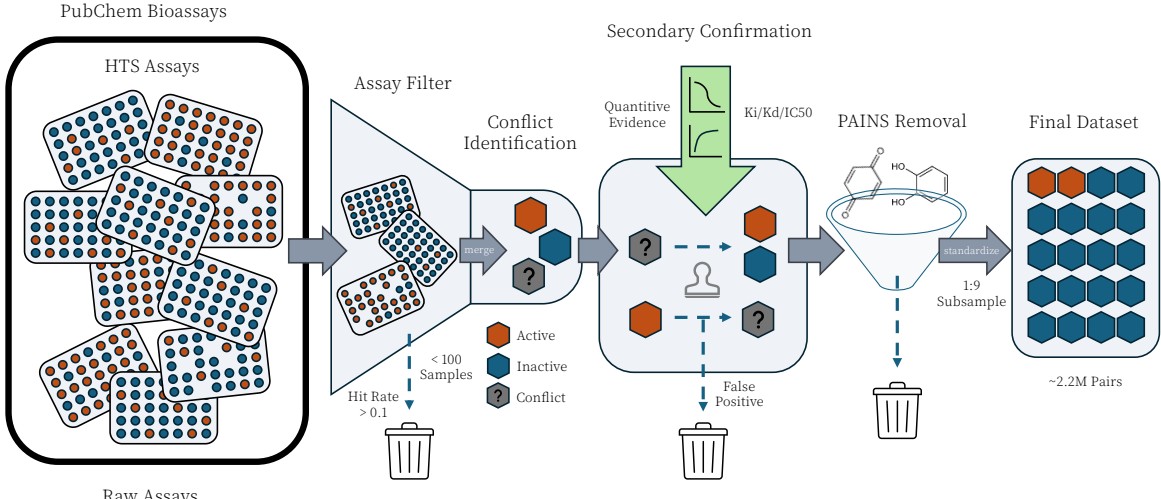

**(b)**

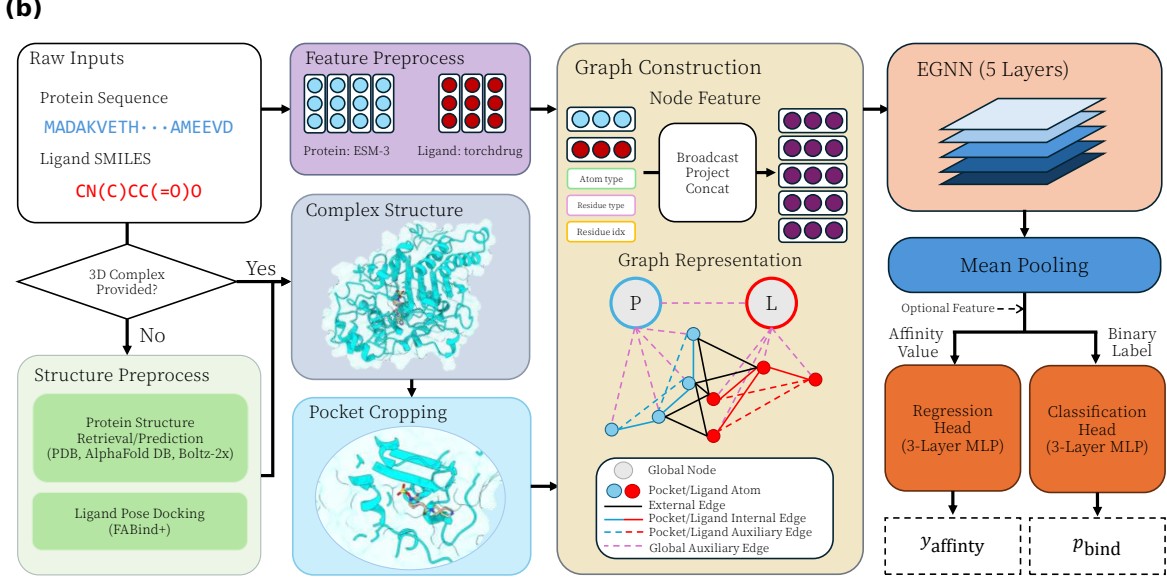

Figure 2: **The FlashBind framework.** **(a)** Training data curation. To mitigate experimental noise, we construct a multi-stage filtration pipeline with assay-consistency checks, secondary confirmation, and PAINS removal. **(b)** Inference pipeline. FlashBind decouples structure generation from scoring: raw inputs are mapped to a 3D complex with FABind+, cropped to the binding interface, and scored by an EGNN to predict the binding probability.

maximize chemical diversity. We then apply a *compound-level* secondary-confirmation step that keeps an active label only when it is corroborated by quantitative evidence ($K_d$, $K_i$, or $IC_{50}$) and discards conflicting entries lacking such confirmation. Finally, Pan-Assay Interference Compounds (PAINS) (Baell & Holloway, 2010) are removed to eliminate frequent hitters, using the full PAINS catalog (the combined PAINS_A/B/C sub-catalogs, 480 substructure filters) as implemented in RDKit's `FilterCatalog`, and a compound is discarded if it matches any filter. After balancing to a 1:9 binder-to-decoy ratio, the final training set contains

**2237058 protein-ligand pairs** spanning **451 protein targets** and **368812 ligands**. For validation we use a curated subset of LIT-PCBA (Tran-Nguyen et al., 2020) comprising 43492 pairs sampled across its 15 targets such that every target retains a hit rate above 0.5%. Of these curation steps, secondary confirmation is the most consequential. An ablation in Appendix E.2 shows that removing it admits unconfirmed and likely false-positive labels and sharply lowers early enrichment (EF@1% from 14.13 to 9.58), so a larger corpus helps only when it is paired with strong noise suppression.

**Enzyme-substrate specificity dataset.** For the fine-grained enzyme-substrate task, we use the ESIBank benchmark (Cui et al., 2025) directly. It comprises **323783 pairs** covering **8124 enzymes** and **34417 ligands**. To ensure a fair comparison, we adhere to the standard "unknown enzyme & substrate" splits defined within the benchmark, matching the protocol of EZSpecificity (Cui et al., 2025).

**Leakage prevention.** To rigorously assess generalization, we enforce strict train/test separation at two levels. At the *protein* level, we cluster all sequences with MMseqs2 (Steinegger & Söding, 2017) and remove any training protein sharing $\geq 90\%$ sequence identity with a validation or test protein. At the *ligand* level, we discard any training ligand whose Tanimoto similarity (Butina, 1999) to an *active* test ligand exceeds 0.4, preventing the model from exploiting memorized scaffolds. Beyond these two controls, we additionally verify assay-level (PubChem AID) disjointness from the test benchmarks, quantify residual Bemis–Murcko scaffold sharing, and probe remote target-family relatedness by re-clustering at 30% sequence identity; the full analysis in Appendix E.1 confirms that our reported performance is not driven by leakage.

### 3.4   Training and Inference

All models are trained on a cluster of NVIDIA L40S GPUs using a group-based mini-batch sampling strategy (Appendix B.4), in which every batch is drawn from a single experimental assay so that the loss compares compounds tested under identical conditions.

Our experiments use three separately trained models: a binary screening classifier trained on the curated PubChem corpus (Section 3.3), which we also apply zero-shot, without any additional training or fine-tuning, to the antibiotic and prospective DnaG tasks (Sections 4.3–4.4); an enzyme-substrate model retrained from scratch on ESIBank (Section 4.2); and an affinity-regression variant trained on SAIR (Appendix A).

**Virtual screening.** To emphasize early enrichment, we optimize a Focal Loss (Lin et al., 2018) ($\gamma = 1$, $\alpha = 0.7$) under a per-assay sampling scheme that enforces a 1:4 binder-to-decoy ratio within each batch.

**Enzyme-substrate specificity.** We follow the EZSpecificity protocol exactly, performing 4-fold cross-validation on the "unknown enzyme & substrate" split with the same Focal Loss. Because catalytic specificity hinges on subtle functional-group chemistry that pure geometry can miss, we augment the geometric representation with UniMol embeddings (Ji et al., 2024) and Morgan fingerprints (Rogers & Hahn, 2010) for this task only, matching the input modalities of the baseline.

**Inference and ensembling.** At inference we ensemble the top $k = 2$ checkpoints selected on validation performance, averaging their outputs for classification. Protein embeddings are computed once per target and amortized across all candidate ligands, and ligand descriptors are generated by lightweight RDKit featurization, so the per-complex cost is dominated by docking and EGNN scoring (analyzed in Section 4.1).

## 4   Experiments

We evaluate FlashBind across three increasingly demanding settings that probe complementary aspects of structure-based prediction. We first establish its core competency in ultra-fast virtual screening on the MF-PCBA benchmark (Buterez et al., 2023), where it attains early enrichment competitive with foundation models (matching Boltz-2 at the top 1%) at a 50-fold lower inference cost, and ablate each design choice to isolate the sources of this efficiency (Section 4.1). We then test generalization beyond thermodynamic binding to enzyme-substrate specificity, a fine-grained functional task governed by catalytic alignment (Section 4.2).

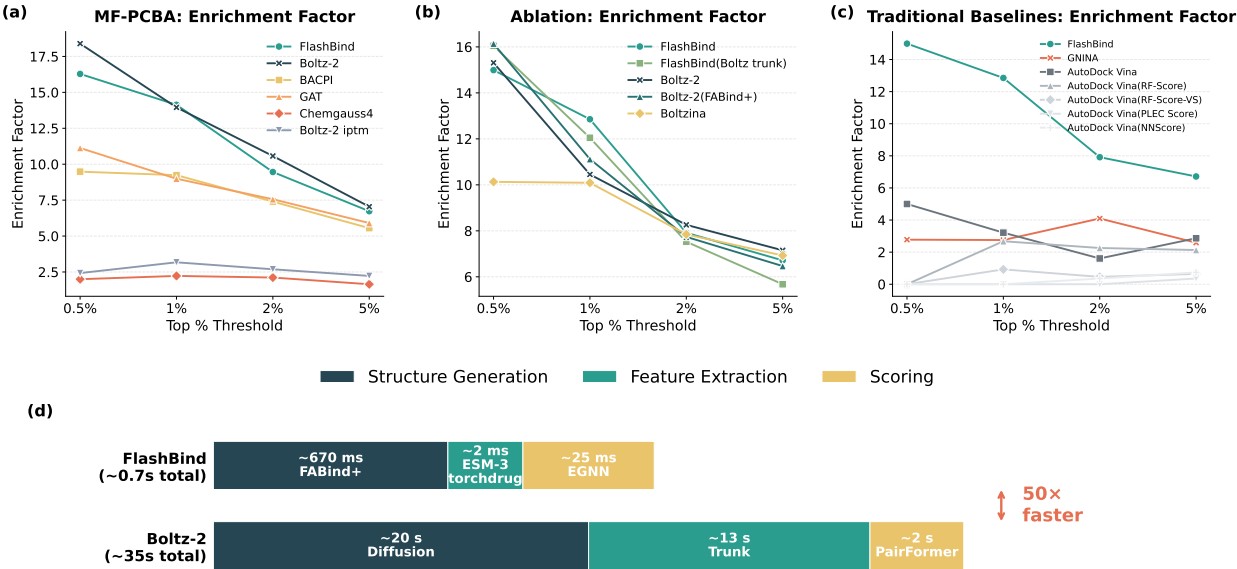

Figure 3: **Enrichment Factor (EF) on the MF-PCBA benchmark and per-complex inference time breakdown.** **(a)** Comparison with deep learning and physics-based scoring baselines on the full benchmark. FlashBind attains early enrichment competitive with the foundation model Boltz-2 while substantially outperforming traditional methods. **(b)** Step-wise ablation study on a representative subset, validating each efficiency-oriented design choice. **(c)** Comparison with traditional docking and rescoring pipelines on the same subset. **(d)** Per-complex inference time breakdown. Each bar is decomposed into three pipeline stages: structure generation, feature extraction, and scoring. FlashBind replaces Boltz-2's diffusion sampling with FABind+, its deep trunk with pre-trained ESM-3/torchdrug embeddings, and its PairFormer scorer with a lightweight EGNN, achieving a cumulative **50-fold speedup** on a single NVIDIA L40S GPU. Bar widths are not to scale. In **(a)** FlashBind is evaluated in-house while all baselines are reproduced from the original Boltz-2 publication; in **(b,c)** all methods are evaluated in-house on a common subset (Appendix F.1).

Finally, we assess real-world utility in antibiotic discovery: FlashBind substantially outperforms both physics-based docking and foundation models on a retrospective *E. coli* benchmark (Section 4.3), and in a fully prospective campaign against DNA primase (DnaG) we experimentally confirm 10 active inhibitors among 136 tested compounds, 4 of which show whole-cell antibacterial activity (Section 4.4).

## 4.1 Ultra-fast Virtual Screening

**Enrichment performance.** To assess the model's capability in identifying active compounds from vast chemical spaces, we benchmarked FlashBind on the MF-PCBA dataset (Buterez et al., 2023), a standard benchmark adopted by Boltz-2 for evaluating virtual screening performance. The primary metric is the enrichment factor (EF), the ability to rank true binders at the very top of a prioritized list.

FlashBind demonstrates exceptional performance in this critical metric. As illustrated in Fig. 3a, our model achieves an Enrichment Factor at the top 1% (EF@1%) of 14.13, significantly outperforming traditional physics-based scoring functions like Chemgauss4 (Nishimoto & Fedorov, 2016) (EF@1% = 2.23) and sequence-based deep learning baselines (Li et al., 2022). Most notably, FlashBind is competitive with the computationally intensive foundation model Boltz-2 (Passaro et al., 2025), which achieves an EF@1% of 13.95 on the same test set. This indicates that our lightweight geometric encoder effectively distills the complex structural signals required for hit identification. These results are stable rather than seed-dependent. Re-running inference under five random seeds leaves every reported metric unchanged (raw scores differ only beyond the fifth decimal place), and the per-target dispersion of FlashBind's enrichment closely matches that of Boltz-2, indicating that the spread reflects cross-target difficulty rather than model instability (Appendix F.3).

**Ablation study.** To rigorously validate the sources of our efficiency and accuracy, we conducted a step-wise ablation study on a representative subset of the benchmark (Fig. 3b), constructed by sampling one-tenth of the compounds per target proportionally to the hit rate, as running ablation variants and traditional docking baselines on the full 500k-compound library is prohibitively expensive. Our analysis supports three key design premises. First, we confirmed that a fast docking oracle provides a sufficient structural foundation. The Boltz-2(FABind+) variant, which utilizes pre-computed poses from FABind+, maintains robust performance compared to the original Boltz-2 (EF@1%: 11.12 vs. 10.45). This conclusion is further supported by Boltzina (Furui & Ohue, 2025), which substitutes the diffusion module with AutoDock Vina (Trott & Olson, 2009) and similarly achieves comparable results (EF@1%: 10.09). Together, these findings suggest that precise conformational sampling from diffusion models is not strictly necessary if a high-quality docked pose is available.

Second, we assessed the necessity of heavy-weight scoring architectures. By comparing Boltz-2(FABind+) with a variant of our model using Boltz-2's latent representations (FlashBind(Boltz trunk)), we observe that replacing the massive PairFormer module with our lightweight EGNN results in no significant performance loss (EF@1%: 12.05 vs. 11.12). This confirms that a streamlined equivariant graph network is sufficient to capture critical protein-ligand interactions.

Third, we validated the use of efficient pre-trained embeddings. The standard FlashBind model, which utilizes accessible ESM-3 (Hayes et al., 2025) and torchdrug (Zhu et al., 2022) features, achieves performance fully comparable to the variant relying on computationally expensive Boltz-2 trunk outputs (EF@1%: 12.85 vs. 12.05), effectively decoupling our framework from the foundation model.

**Comparison with traditional pipelines.** Beyond internal validation, we benchmarked FlashBind against established traditional docking and rescoring methods on the same subset (Fig. 3c). Our geometric deep learning approach substantially outperforms standard AutoDock Vina scoring (EF@1% = 3.21) as well as random forest and neural network-based rescoring functions (e.g., RF-Score, GNINA (Ballester & Mitchell, 2010; Wójcikowski et al., 2017; 2019; Durrant & McCammon, 2010; McNutt et al., 2021)).

**Inference efficiency.** The cumulative effect of these architectural optimizations is a decisive improvement in practical throughput, as visualized by the per-complex time breakdown in Fig. 3d. For Boltz-2, each complex requires $\sim$35 s of wall-clock time: $\sim$20 s for iterative diffusion-based structure generation, $\sim$13 s for computing trunk representations through the deep PairFormer, and $\sim$2 s for final scoring. FlashBind systematically replaces every expensive component with a lightweight counterpart. Acquiring the target's protein structure (retrieved from the PDB or AlphaFold DB, or predicted with Boltz-2x only as a last resort) and computing its ESM-3 embedding are *one-time, per-target* operations. Structure acquisition takes $\sim$8.2 s per target; amortized over a representative library of 5000 ligands screened against that target, it contributes only $\sim$1.6 $\times$ $10^{-3}$ s per complex, and the ESM-3 embedding ($\sim$0.2 s per target) is smaller still, so both are negligible against the docking-dominated cost. The per-complex steps are likewise streamlined: FABind+ generates a docked pose in $\sim$0.67 s; torchdrug molecular features, based on lightweight RDKit descriptors, are generated at a throughput exceeding 800 molecules per CPU-second (negligible per complex); and the equivariant EGNN scores each complex in only $\sim$25 ms (Table 4). Summing every contribution, the end-to-end latency is $\sim$0.7 s, a **50-fold speedup**, with the docking oracle as the sole remaining bottleneck. In a practical rescoring scenario where a pre-docked pose library is already available, the scoring step alone enables evaluation of over 140000 complexes per GPU-hour, making million-scale campaigns tractable on modest hardware. Fig. 1a further contextualizes this gain as a Pareto frontier of enrichment versus inference cost: FlashBind occupies the optimal region, matching Boltz-2's top-1% early enrichment at a fraction of the computational budget. A complete per-step breakdown, the measurement protocol, the per-target amortization analysis, and a library-scale projection are provided in Appendix D.

### 4.2 Enzyme-Substrate Interaction

In addition to virtual screening, another application of the protein-ligand binding predictor is the decoding enzyme-substrate specificity, a functional property governed by precise catalytic alignment rather than thermodynamic stability alone. To evaluate FlashBind's ability in this fine-grained regime, we conducted a

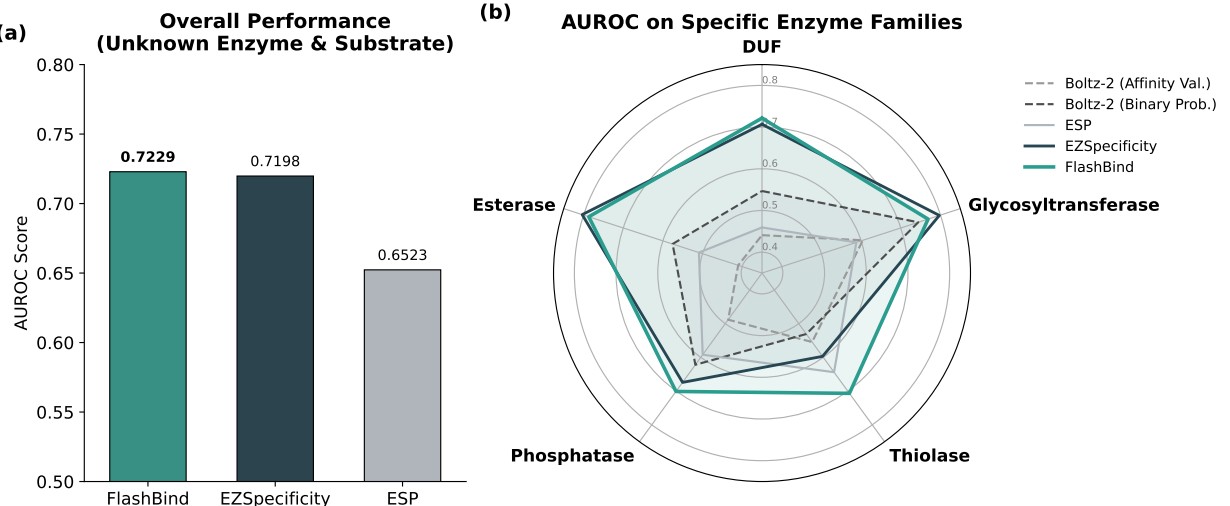

Figure 4: **Evaluation of enzyme-substrate specificity prediction on the ESIBank benchmark.** **(a)** Overall performance comparison on the "unknown enzyme & substrate" split (4-fold cross-validation). FlashBind achieves an AUROC of 0.7229, performing comparably to the specialized state-of-the-art model EZSpecificity (0.7198) and significantly outperforming sequence-based baselines like ESP. **(b)** Fine-grained analysis on representative enzyme families (Thiolase, Glycosyltransferase, DUF, Phosphatase, Esterase). FlashBind demonstrates robust generalization in these specific categories, outperforming the foundation model Boltz-2 and matching the specialized architecture of EZSpecificity, indicating that our geometric encoder effectively captures functional catalytic patterns. FlashBind and Boltz-2 are evaluated in-house on the identical benchmark split and metric; EZSpecificity and ESP are reproduced from prior work (Appendix F.1).

case study using the ESIBank data set (Cui et al., 2025), a comprehensive benchmark for enzyme specificity prediction, on which FlashBind was retrained from scratch.

We rigorously evaluated our model under the "unknown enzyme & substrate" split, the most challenging setting, where neither the protein nor the small molecule have been seen during training. Following the protocol of the state-of-the-art method EZSpecificity (Cui et al., 2025), we performed 4-fold cross-validation to ensure statistical robustness. Recognizing that enzyme specificity is often governed by subtle chemical functional group interactions that pure geometric scoring might miss, we adopted a similar strategy to the EZSpecificity framework by augmenting our geometric encoder with explicit chemical descriptors (UniMol embeddings (Ji et al., 2024) and Morgan fingerprints (Rogers & Hahn, 2010)).

As shown in Fig. 4a, FlashBind achieves an overall AUROC of 0.7229, demonstrating performance fully comparable to the specialized EZSpecificity model (AUROC = 0.7198) and significantly outperforming the sequence-based baseline ESP (Kroll et al., 2023) (AUROC = 0.6523). This result is particularly notable given that FlashBind utilizes a general-purpose geometric encoder, whereas EZSpecificity employs a heavy, task-specific cross-attention architecture designed exclusively for this problem.

We further investigated performance across specific enzyme families, including data-scarce categories like Thiolases and Domain of Unknown Function (DUF) proteins (Fig. 4b). In these specific regimes, FlashBind consistently outperforms the foundation model Boltz-2 and the sequence-based ESP, while maintaining parity with EZSpecificity. For instance, in the Glycosyltransferase family, our model effectively captures the subtle structural determinants required for sugar transfer, a task where pure sequence-based methods often falter. The ability to match a specialized SOTA model on its own benchmark serves as strong validation of our architectural soundness, suggesting that the geometric features learned by FlashBind are not limited to binding affinity but are transferable to complex functional prediction tasks.

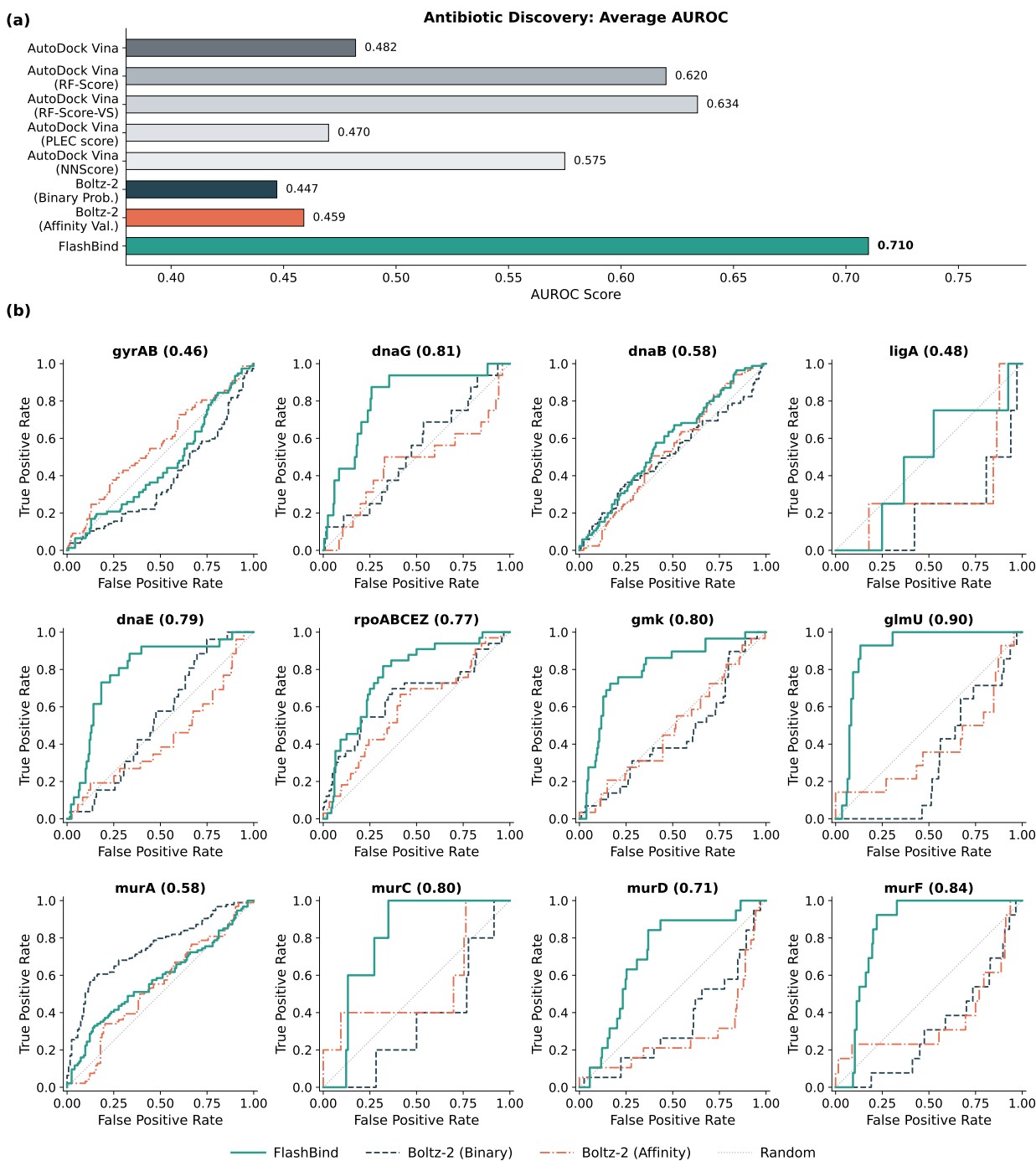

Figure 5: **Benchmarking performance on the antibiotic discovery task. (a)** Comparison of average AUROC across 12 essential *E. coli* targets. FlashBind (0.710) demonstrates practical ranking capability, significantly outperforming standard docking (AutoDock Vina) and machine-learning rescoring functions. **(b)** Representative ROC curves for key targets including guanylate kinase (*gmk*) and bifunctional acetyltransferase (*glmU*). FlashBind (green) maintains higher true positive rates compared to baselines. FlashBind and Boltz-2 are evaluated in-house on the identical benchmark test set and metric; AutoDock Vina and the ML rescoring baselines are reproduced from prior work (Appendix F.1).

### 4.3 Antibiotic Discovery Benchmark

While the previous section established FlashBind as an ultra-fast filter for large-scale screening benchmarks, validating its utility in real-world discovery campaigns is the ultimate test. To this end, we apply FlashBind to a structure-based antibiotic discovery task (Wong et al., 2022). This task is particularly challenging, as a model must not only identify compounds that bind bacterial targets but also inhibit bacterial growth, and must do so within a chemical space distinct from standard training sets.

The benchmark dataset (Wong et al., 2022) comprises 218 active antibacterial compounds and 100 inactive compounds docked to essential *E. coli* proteins, whose structures are predicted by AlphaFold2. Ground-truth labels are derived from *in vitro* enzymatic inhibition assays across 12 essential proteins of *E. coli* (e.g., DNA gyrase, MurA). A compound is labeled positive if it shows more than 50% inhibition in both replicates. Model performance is evaluated via the area under the receiver operating characteristic curve (AUROC), averaged across all 12 target proteins.

We compared FlashBind with Boltz-2, standard molecular docking tools such as AutoDock Vina (Trott & Olson, 2009), and various machine-learning scoring functions (Fig. 5). To evaluate the zero-shot performance of FlashBind, we tested all models without finetuning. Concretely, FlashBind here is the same PubChem-trained screening classifier of Section 3.4, applied directly to this distinct chemical and target space with no additional training on antibiotic data. We found that traditional physics-based docking struggles to distinguish actives from decoys (Vina Avg. AUROC $\approx 0.48$). Similarly, Boltz-2 does not transfer effectively to this dataset, with performance comparable to random guessing (AUROC $\approx 0.45$).

In contrast, FlashBind demonstrates practical utility in this regime, achieving a mean AUROC of 0.710. As shown in the target-specific ROC curves (Fig. 5b), our model consistently retrieves active scaffolds for critical targets such as *gmk* and *glmU*. The performance gap compared to baselines suggests that our approach is sufficiently robust to prioritize candidates even in complex biological assays, making it a viable tool for the initial stages of antibiotic discovery campaigns.

Furthermore, as illustrated in the Pareto frontier analysis (Fig. 1b), FlashBind lies above the Pareto front, achieving superior accuracy while remaining over $6\times$ faster than AutoDock Vina and other rescore method (0.7s vs. 4.5s per complex). This positions FlashBind as the only method simultaneously surpassing both the accuracy and efficiency of existing baselines in this benchmark.

### 4.4 Prospective Wet-Lab Validation

To assess whether FlashBind's predictions translate into real-world success, we conducted a prospective virtual screening campaign targeting *E. coli* DNA primase (dnaG). We scored all 9289 compounds from the Broad Institute compound collection using FlashBind (Fig. 6a). From the top-ranked candidates, we applied two filters to ensure chemical quality and diversity: compounds matching the same PAINS catalog used during training (Section 3.3) were excluded, and redundant scaffolds were removed by retaining only compounds with pairwise Tanimoto similarity below 0.5. This yielded a final selection of 136 compounds for experimental testing.

Following Wong et al. (Wong et al., 2022), each compound was assayed against DnaG in two independent replicates at a concentration of $100\mu$M. Full assay conditions are provided in Appendix C. Normalized inhibition scores were averaged to produce a final activity score, where higher scores indicate stronger DnaG inhibition, and a compound was classified as active if it induced more than 50% inhibition of DnaG. As shown in Fig. 6a, 10 of the 136 compounds tested were confirmed to be active, corresponding to a hit rate of 7.4%. This substantially exceeds the typical hit rates observed in unguided random screening campaigns, demonstrating that FlashBind's rankings provide actionable enrichment in a prospective setting.

We further evaluated each dnaG inhibitor for whole-cell antibacterial activity using two *E. coli* inhibition assays at a concentration of $128\mu$g/mL (Fig. 6b). The first assay measures *E. coli* cell viability upon compound treatment alone, while the second assay co-administers a sub-inhibitory concentration of polymyxin B (PMB) to permeabilize the outer membrane. The goal of the second assay is to isolate intracellular target inhibition from membrane permeability, since the screening process did not take into account permeability.

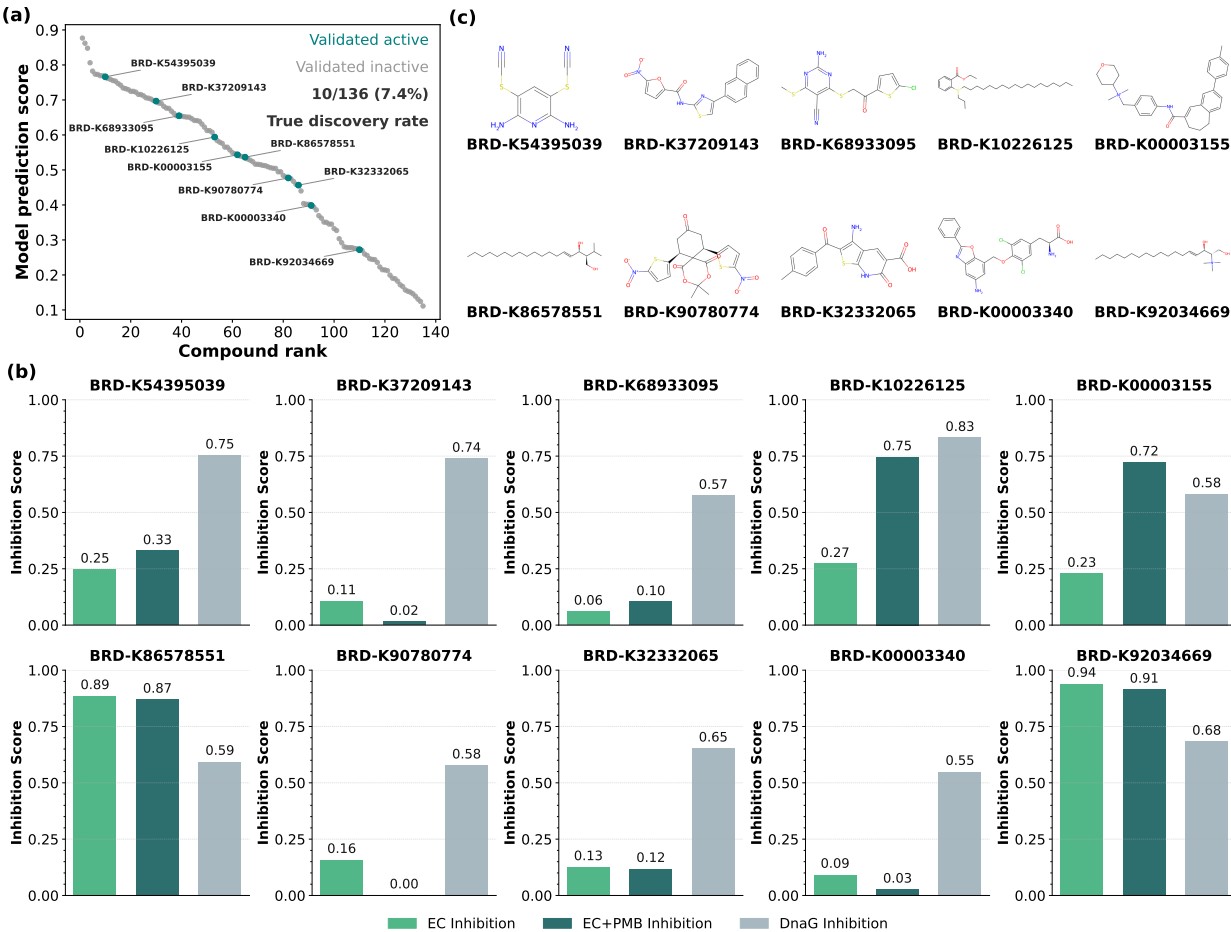

Figure 6: **Prospective wet-lab validation of FlashBind on DnaG. (a)** Rank-ordered FlashBind prediction scores for the 136 selected compounds screened against *E. coli* DNA primase (dnaG). Teal points indicate experimentally confirmed actives (DnaG inhibition ≥ 50%), yielding a hit rate of 10/136 (7.4%).**(b)** Experimental profiles of all 10 confirmed hits, showing EC inhibition, EC+PMB inhibition, and DnaG inhibition for each compound. Higher values indicate stronger inhibitory activity. **(c)** Chemical structures of all 10 confirmed hits identified by FlashBind.

Among the 10 confirmed hits, 4 compounds exhibited strong EC+PMB inhibition (≥ 50%) alongside high DnaG inhibition, demonstrating that FlashBind can effectively prioritize compounds with genuine whole-cell antibacterial potential. All 10 confirmed hits span structurally diverse chemotypes (Fig. 6c); for example, BRD-K00003155 combines potent DnaG inhibition with whole-cell activity within a compact, drug-like scaffold, illustrating that FlashBind captures genuine binding signals rather than overfitting to a narrow chemical series.

## 5 Discussion and Conclusion

We presented FlashBind, a lightweight geometric framework that resolves the long-standing efficiency-accuracy trade-off in structure-based virtual screening. By occupying the optimal region of the Pareto frontier, it matches the top-1% early enrichment of large-scale foundation models such as Boltz-2 with a 50-fold reduction in inference latency, making structure-based screening of massive libraries accessible to laboratories without immense computational resources.

Our findings challenge the assumption that heavy-weight, end-to-end structure generation is a prerequisite for high-accuracy scoring. The success of FlashBind indicates that precise conformational sampling via expensive diffusion models is not strictly necessary for hit identification: a lightweight $E(3)$-equivariant encoder can capture the protein-ligand interactions present in fast docking priors such as FABind+ (Gao et al., 2025), relying on robust geometric features like heavy-atom contact patterns rather than precise atomic coordinates that demand explicit hydrogen or solvent modeling. Equally, scale alone is insufficient: our expanded training set was only effective when coupled with rigorous filtration to suppress experimental noise, suggesting that data quantity and quality control must be pursued together.

This robustness extends to other biological tasks. In the antibiotic discovery and enzyme specificity campaigns, where foundation models frequently suffered from negative transfer between stable crystal structures and complex enzymatic assays, FlashBind maintained high predictive validity; by prioritizing explicit local geometric constraints, it mitigates overfitting to the protein families and eukaryotic targets that dominate standard training sets. The same task-agnostic encoder also extends to affinity regression (Appendix A), where it surpasses traditional baselines but, we acknowledge, does not consistently match Boltz-2 (Passaro et al., 2025) on absolute affinity prediction; this remains a genuine limitation of the current model. Our encoder ablation (Appendix A.4) shows that replacing the EGNN with a heavier PairFormer yields negligible improvement, which lets us reasonably rule out limited architectural capacity as the primary cause. We therefore hypothesize that the gap is driven mainly by training-data provenance, in particular Boltz-2's larger and more rigorously curated proprietary affinity corpus; we present this as an empirical inference rather than a definitive conclusion, since other factors we did not fully control, such as differences in input preprocessing or labeling, may also contribute.

The modular design underlying FlashBind's efficiency also defines its principal limitation. Its predictive ceiling is bounded by the fidelity of the upstream docking oracle. Where FABind+ fails to produce a plausible pose, under significant conformational plasticity or at cryptic binding sites, the downstream EGNN (Satorras et al., 2022) may process incorrect geometric signals. Although ground-truth poses are unavailable on these benchmarks and failures thus cannot be verified directly, FlashBind's accuracy visibly tracks the definability of the binding site. This is clearest on the antibiotic benchmark (Fig. 5). The strongest targets are metabolic enzymes with well-defined small-molecule pockets (*glmU*, AUROC 0.90; *murF*, 0.84; *gmk* and *murC*, 0.80), whereas the weakest are large nucleic-acid-processing enzymes that act on extended interfaces and are difficult substrates for conventional docking (DNA gyrase *gyrA*/*gyrB*, 0.46; DNA ligase *ligA*, 0.48; the replicative helicase *dnaB*, 0.58). The same trend holds on MF-PCBA, where the weakest targets are shallow protein-interaction or regulatory surfaces rather than classical druggable pockets, such as a protein-CTD phosphatase (CTDSP1, 0.68) and an E2 ubiquitin-conjugating enzyme (UBE2N, 0.60). Errors thus concentrate in targets that lack a clearly defined pocket, the regime where pocket-based docking is least reliable. This is consistent with our hypothesis that upstream pose error propagates to the scorer, though these remain correlational observations across targets rather than a controlled mechanistic attribution. Future work will reduce this dependency via lightweight flexible-docking modules and more expressive geometric networks that capture higher-order many-body interactions over broader bioactivity datasets. Taken together, FlashBind offers a scalable, accurate, and physically grounded framework that redefines the practical limits of structure-based virtual screening.

### Broader Impact Statement

FlashBind accelerates structure-based virtual screening, which can speed the discovery of beneficial therapeutics, including new antibacterials against drug-resistant pathogens. We also recognize the dual-use nature of bioactivity-prediction models. A system that ranks compounds by predicted target binding could in principle be redirected toward harmful ends, such as prioritizing toxic agents. Several factors bound this risk in the present case. FlashBind is a scoring-and-ranking model that operates over pre-existing chemical libraries, not a generative system that designs novel molecules *de novo*, so its outputs are constrained to the libraries to which it is applied. Moreover, any predicted activity must still be realized through a resolved target structure, a synthesizable and synthesized compound, and experimental confirmation; computational scores are not validated therapeutic candidates absent rigorous wet-lab verification, and should not be overinterpreted as

such. We therefore regard the responsible-use considerations for FlashBind as standard for virtual-screening tools, and we encourage the use of appropriate access and screening safeguards when deploying such models.

**Acknowledgements**

To support reproducibility, we will publicly release the complete FlashBind pipeline, including the curated dataset splits, the model and training code, trained checkpoints, and the evaluation scripts, so that all results in this paper can be reproduced end to end. To preserve anonymity during review we omit repository links here, and will provide them, together with a public license, in the camera-ready version. Full implementation details, node featurization, pocket cropping, edge construction, group-based sampling, and optimizer settings, together with the complete wet-lab protocols, are already documented in Appendices B and C.

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

# A    Binding Affinity Prediction

While the main text focuses on the binary classification task for high-throughput virtual screening, the geometric encoder of FlashBind is inherently task-agnostic. To adapt the framework for quantitative binding affinity prediction (regression), we replaced the binary classification head with an architecturally identical Multi-Layer Perceptron (MLP) regression head, simply removing the final sigmoid activation function to produce a continuous scalar output. In this section, we demonstrate the model's capability in this regime and investigate the impact of encoder architecture on performance.

## A.1    Data Curation and Filtering

To train the model for continuous affinity prediction (e.g., $K_i, K_d, IC_{50}$), we utilized the **SAIR** dataset (Lemos et al., 2025), a large-scale synthetic structural dataset integrating data from ChEMBL (Zdrazil et al., 2023) and BindingDB (Liu et al., 2006). Our curated training set comprises 796557 protein-ligand pairs, covering 28821 proteins and 403124 unique ligands.

Unlike the binary classification task where assays were merged to maximize chemical diversity, for the regression task, we strictly grouped entries by their unique assay identifier defined by the combination of `protein` (UniProt ID) (Consortium, 2024), `source` (ChEMBL or BindingDB), and `description`. This ensures that all data points within a group originate from a single assay with a consistent experimental setup, as continuous affinity values are highly dependent on specific experimental contexts.

We applied a rigorous filtration pipeline focusing on both label reliability and structural quality:

**Label Quality Filtering**    Following the protocol of Boltz-2, we applied the following filters to each assay group:

- All affinity values were standardized to a common logarithmic scale relative to a 1 $\mu$M baseline.

- We discarded assays where the mean pairwise Tanimoto similarity (using ECFP4 Morgan fingerprints) (Rogers & Hahn, 2010) between compounds was below 0.25. Such assays lack a discernible chemical series, making them uninformative for learning structure-activity relationships.

- We removed assays with fewer than 10 data points, fewer than 10 unique affinity values, or a unique-to-total ratio below 0.2.

- We discarded assays where the standard deviation of internal affinity values was below 0.25 (log scale), as a narrow activity range provides insufficient signal for regression.

- Assays containing extreme affinity values (less than $10^{-6}$ $\mu$M) were discarded to prevent artifacts from unit inconsistencies.

**Structural Quality Filtering**    For each protein-ligand complex, SAIR provides candidate structures predicted by Boltz-1x (Wohlwend et al., 2024). We filtered individual data points based on structural fidelity:

- We discarded entries where the mean ipTM of the predicted structure was $\leq 0.5$, indicating low confidence in the binding interface.

- We discarded entries where the PoseBusters (Buttenschoen et al., 2023) pass rate across the five predicted poses was $\leq 0.5$ (i.e., fewer than three poses passed consistency checks).

The threshold of 0.5 was chosen as a deliberate trade-off between rigor and data retention, ensuring that the retained structures are physically plausible while maintaining a dataset scale sufficient for training.

Table 1: **Comprehensive performance comparison on affinity value prediction benchmarks.** Metrics are averaged per-assay. "non-cent." and "cent." denote metrics computed on raw and centered predictions, respectively. PW1/PW2 refer to the percentage of predictions within 1 and 2 kcal/mol of the experimental value. [†] Values reproduced from prior work (baselines sourced from Boltz-2); the unmarked FlashBind rows are evaluated in-house (Appendix F.1).

| Dataset | Method | Pearson R ↑ | Kendall $\tau$ ↑ | PMAE ↓ | MAE ↓ | | PW1 ↑ | | PW2 ↑ | |
|---|---|---|---|---|---|---|---|---|---|---|
| | | | | | non-cent. | cent. | non-cent. | cent. | non-cent. | cent. |
| OpenFE | Boltz-2[†] | 0.62 | 0.46 | 0.93 | 1.22 | 0.64 | 0.49 | 0.80 | 0.82 | 0.96 |
| | BACPI[†] | 0.29 | 0.19 | 1.21 | 1.44 | 0.85 | 0.40 | 0.67 | 0.74 | 0.94 |
| | GAT[†] | 0.28 | 0.20 | 1.30 | 1.42 | 0.91 | 0.40 | 0.64 | 0.75 | 0.92 |
| | FlashBind | 0.44 | 0.33 | 1.13 | 1.46 | 0.79 | 0.39 | 0.71 | 0.71 | 0.95 |
| FEP+ (4 targets) | Boltz-2[†] | 0.66 | 0.48 | 0.85 | 0.75 | 0.59 | 0.69 | 0.83 | 0.97 | 0.98 |
| | BACPI[†] | 0.14 | 0.09 | 1.18 | 1.40 | 0.82 | 0.43 | 0.62 | 0.73 | 1.00 |
| | GAT[†] | 0.40 | 0.28 | 1.07 | 1.19 | 0.71 | 0.43 | 0.72 | 0.86 | 0.95 |
| | FlashBind | 0.53 | 0.38 | 1.10 | 1.44 | 0.76 | 0.39 | 0.71 | 0.74 | 0.95 |
| CASP16 | Boltz-2[†] | 0.65 | 0.45 | 1.36 | 1.28 | 0.95 | 0.48 | 0.61 | 0.81 | 0.90 |
| | BACPI[†] | 0.41 | 0.31 | 1.55 | 1.25 | 1.10 | 0.45 | 0.51 | 0.81 | 0.89 |
| | GAT[†] | 0.50 | 0.35 | 1.58 | 1.28 | 1.13 | 0.44 | 0.49 | 0.79 | 0.84 |
| | FlashBind | 0.65 | 0.51 | 1.14 | 1.21 | 0.88 | 0.29 | 0.68 | 0.94 | 0.94 |

**Validation and Test Sets** To prevent data leakage, we removed any protein from the training set sharing $\geq 90\%$ sequence identity with proteins in the validation or test sets, and any ligand with a Tanimoto similarity $> 0.4$ to active ligands in the test set.

- **Validation**: Constructed by randomly holding out all data points from 20 diverse assays in the SAIR dataset.

- **Test**: We evaluated performance on three benchmarks identical to those used in Boltz-2: the **OpenFE subset** (Gowers et al., 2023), a **4-target subset (CDK2, TYK2, JNK1, P38)** of the FEP+ benchmark (Hahn et al., 2022), and the blind **CASP16** benchmark (Gilson et al., 2025).

## A.2 Training Objective and Inference

**Composite ranking objective.** Because binding affinity is reported on assay-specific scales, absolute values are far less transferable across experiments than the *relative* ordering of compounds within a single assay. We therefore train the regression head with a composite objective that weights pairwise ranking accuracy over absolute error (Huber, 1992):

$$\mathcal{L}_{\text{affinity}} = 0.9 \cdot \mathcal{L}_{\text{Huber}}(y_i - y_j, \, \hat{y}_i - \hat{y}_j) + 0.1 \cdot \mathcal{L}_{\text{Huber}}(y_i, \, \hat{y}_i), \tag{1}$$

where $(i, j)$ index two compounds drawn from the same assay group (Appendix B.4). The first term penalizes errors in the predicted pairwise difference and dominates the loss, while the second term anchors the predictions to the absolute scale. This emphasis on intra-assay ranking aligns the training signal with the lead-optimization use case, where prioritizing compounds within a series matters more than exact $K_d$ recovery.

**Molecular-weight bias correction.** Structure-based scoring functions are known to exhibit a systematic bias toward larger ligands, as additional atoms tend to inflate raw interaction scores irrespective of true binding strength. To mitigate this, we apply a post-hoc polynomial correction to the ensemble-averaged predictions as a function of ligand molecular weight (MW), fitted on the validation set and applied at inference. This calibration removes size-correlated artifacts without retraining and is applied only to the regression task; the classification setting, which operates on within-assay ranking of $p_{\text{bind}}$, does not use it.

### A.3 Benchmarking Results

We compared our method FlashBind against the state-of-the-art foundation model Boltz-2, as well as sequence-based (BACPI) (Li et al., 2022) and ligand-only (GAT) baselines.

The comprehensive results are summarized in Table 1. FlashBind consistently and significantly outperforms the non-structural baselines across all datasets. For instance, on the FEP+ 4-target benchmark, our model achieves a Pearson's R of 0.53, a substantial improvement over the sequence-based BACPI (0.14). This performance gap underscores the effectiveness of our geometric encoder in leveraging 3D structural information for affinity prediction.

When compared to the computationally intensive Boltz-2 model, our lightweight approach achieves competitive performance, particularly in ranking metrics. This is most evident on the blind CASP16 benchmark, where FlashBind's rank correlation matches and slightly exceeds that of Boltz-2 (Kendall's $\tau$ of 0.51 vs. 0.45), highlighting its strong generalization capability in challenging, blind evaluation settings. While Boltz-2 generally yields lower absolute errors on the OpenFE and FEP+ benchmarks, FlashBind's performance remains comparable, validating its utility as an efficient alternative for rapid affinity estimation.

### A.4 Ablation of Encoder Architecture

To further investigate the performance gap between FlashBind and Boltz-2 observed in the affinity prediction benchmarks, we conducted an additional ablation study focusing on the model's encoder capacity. A potential hypothesis for the performance difference is that the $E(3)$-equivariant Graph Neural Network (EGNN) (Satorras et al., 2022) used in FlashBind might lack the representational power of the computationally heavier PairFormer architecture employed by Boltz-2.

To test this hypothesis, we developed a variant of our model, denoted as **FlashBind (PairFormer)**. In this variant, we replaced the EGNN encoder with a PairFormer module with hyperparameters similar to the Boltz-2 affinity head, while keeping the rest of the pipeline identical (i.e., using FABind+ generated structures as input and the same MLP prediction head).

The results are summarized in Table 2. Contrary to the expectation that a more complex architecture would yield significant gains, the PairFormer variant demonstrated negligible performance improvements compared to the standard EGNN-based FlashBind. For instance, on the OpenFE dataset, the Pearson's R only marginally fluctuated (from 0.44 to 0.43), and on the FEP+ 4 benchmark, the performance remained statistically comparable.

Consequently, this result validates our architectural choice: the EGNN provides a much more favorable trade-off, offering comparable accuracy to a Transformer-based architecture at a fraction of the computational and memory cost. Furthermore, this finding implies that the remaining performance gap between FlashBind and Boltz-2 on affinity regression tasks is unlikely to originate from architectural differences. A more plausible explanation lies in the disparity of training data: Boltz-2 is trained on a proprietary, rigorously curated dataset that is approximately 1.5x larger than our training set derived from SAIR (Lemos et al., 2025), and likely contains substantially fewer noisy labels. This data advantage, rather than the sophistication of the scoring network, is the more probable driver of Boltz-2's stronger affinity prediction performance.

## B Details about the Methods

This section supplements the "Methods" section by providing specific hyperparameters, feature definitions, and algorithmic logic used for graph construction and model training.

### B.1 Node Featurization

To construct the node representations for the EGNN, we generate initial features for proteins and ligands using distinct pipelines before projecting them into a unified space.

Table 2: **Ablation study on encoder architecture.** Detailed performance comparison between the proposed EGNN-based scoring module and a computationally heavier PairFormer-based variant across three benchmarks. The results indicate that increasing the encoder complexity yields negligible performance gains. Both variants are evaluated in-house under an identical protocol.

| Dataset | Method | Pearson R ↑ | Kendall tau ↑ | PMAE ↓ | MAE ↓ | | PW1 ↑ | | PW2 ↑ | |
|---|---|---|---|---|---|---|---|---|---|---|
| | | | | | non-cent. | cent. | non-cent. | cent. | non-cent. | cent. |
| OpenFE | FlashBind (EGNN) | 0.44 | 0.33 | 1.13 | 1.46 | 0.79 | 0.39 | 0.71 | 0.71 | 0.95 |
| | FlashBind (PairFormer) | 0.43 | 0.32 | 1.11 | 0.84 | 0.76 | 0.32 | 0.61 | 0.74 | 0.95 |
| FEP+ 4 targets | FlashBind (EGNN) | 0.53 | 0.38 | 1.10 | 1.44 | 0.76 | 0.39 | 0.71 | 0.74 | 0.95 |
| | FlashBind (PairFormer) | 0.53 | 0.39 | 1.13 | 1.39 | 0.78 | 0.47 | 0.68 | 0.79 | 0.97 |
| CASP16 | FlashBind (EGNN) | 0.65 | 0.51 | 1.14 | 1.21 | 0.88 | 0.29 | 0.68 | 0.94 | 0.94 |
| | FlashBind (PairFormer) | 0.71 | 0.53 | 1.19 | 1.28 | 0.89 | 0.68 | 0.71 | 0.94 | 0.97 |

**Protein Features**  We utilize the **ESM-3** (Hayes et al., 2025) language model (`esm3_sm_open_v1`) to capture sequence-based semantics. We extract the representation from the final hidden layer for each residue. To map these residue-level embeddings to the atomic graph, we broadcast the embedding of a residue to all its constituent atoms. These embeddings serve as the initial sequence features and are computationally efficient to generate, given the limited number of unique protein targets in screening tasks.

**Ligand Features**  We adopt a featurization pipeline same to **torchdrug** (Zhu et al., 2022) using **RD-Kit** (Landrum et al., 2025). For each ligand atom, we extract a feature vector consisting of the following one-hot encoded chemical properties:

- Atom symbol (e.g., C, N, O, S, F, Cl, etc.).

- Atom degree (number of heavy-atom neighbors).

- Total number of attached hydrogen atoms.

- Implicit valence.

- Formal charge.

- Aromaticity (boolean flag).

This pipeline is highly optimized for CPU execution, processing over 800 ligands per second, which allows for on-the-fly feature generation during training.

**Unified Representation**  Protein and ligand features are projected into a common hidden dimension via separate linear layers and concatenated to form a base embedding. This base embedding is further augmented by concatenating: (1) One-hot encodings for atom type and residue type; (2) A binary indicator for molecule type (protein vs. ligand); (3) Positional encodings representing the residue's sequence index.

### B.2  Pocket Cropping

We apply a deterministic, budget-constrained cropping algorithm to isolate the binding interface. The algorithm operates in a greedy manner to select a subset of protein residues $\mathcal{P}$ based on spatial proximity to the ligand, subject to an atom budget $B_a = 2048$ and a residue budget $B_r = 512$.

The procedure is as follows:

1. **Initialization**: Calculate the effective budgets for the protein by subtracting the number of ligand atoms: $B'_a = B_a - |\mathcal{A}_L|$ and $B'_r = B_r - |\mathcal{A}_L|$.

2. **Distance Calculation**: For every protein residue, calculate the minimum Euclidean distance between any of its atoms and ligand center.

3. **Sorting**: Sort all protein residues in ascending order of this distance.

4. **Greedy Selection**: Iterate through the sorted list. Add a residue to the pocket candidates $\mathcal{P}_{budget}$ if adding it does not exceed $B'_a$ or $B'_r$.

5. **Distance Filtering**: Filter $\mathcal{P}_{budget}$ to retain only residues within a cutoff $d_{\max} = 20.0$ Å, forming the set $\mathcal{P}_{dist}$.

6. **Robustness Fallback**: If $|\mathcal{P}_{dist}| < k_{\min}$ (where $k_{\min} = 100$), ignore the distance cutoff and return the top $k_{\min}$ closest residues from $\mathcal{P}_{budget}$ to prevent creating overly sparse graphs.

## B.3 Edge Construction

Inspired by MEAN (Kong et al., 2023), edges are constructed to capture interactions at multiple scales. All edges are directed. If the total number of edges exceeds the budget $B_e = 16384$, the edge list is truncated based on the priority scheme described below.

**Edge Categories**

- **Internal Edges ($\mathcal{E}_{\mathbf{internal}}$)**:
    - *Ligand Covalent*: Single, double, triple, and aromatic bonds derived from the molecular graph.
    - *Protein Covalent*: Intra-residue bonds based on standard amino acid templates.
    - *Protein Sequential*: Connections between C$\alpha$ atoms of adjacent residues ($k$ to $k+1$).

- **External Edges ($\mathcal{E}_{\mathbf{external}}$)**:
    - *Protein-Ligand Proximity*: Connected if Euclidean distance $\|\mathbf{x}_i - \mathbf{x}_j\| < d_{\mathrm{cross}}$ (10.0 Å).

- **Auxiliary Edges ($\mathcal{E}_{\mathbf{aux}}$)**:
    - *Global*: Connections from global virtual nodes ($v_{\mathrm{P}}, v_{\mathrm{L}}$) to all respective atoms, and between $v_{\mathrm{P}}$ and $v_{\mathrm{L}}$.
    - *Protein Spatial*: Connections between atoms of different residues if distance $< d_{\mathrm{protein}}$ (4.0 Å).
    - *Ligand LAS (Local Atomic Structure)*: Virtual edges between ligand atoms within a 2-hop covalent distance or within the same ring.

**Priority Scheme** To ensure critical interactions are preserved under the edge budget, edges are added in the following strict priority order (highest to lowest):

External (Protein-Ligand) > Global > Ligand Covalent > Protein Sequential > Protein Covalent > Ligand LAS > Protein Spatial.

## B.4 Group-Based Sampling

To optimize training stability across diverse datasets, we employ a group-based mini-batch sampling strategy. Each training batch ($B = 20$) consists of multiple independent groups ($N = 5$ samples per group), where all samples in a group originate from the same experimental assay.

The sampling procedure is performed as follows:

1. **Assay Filtering**: Prior to training, we discard invalid assays. Binary assays must contain at least one binder and one decoy. Affinity assays must contain at least two ligands.

2. **Dataset Selection**: For each batch, a dataset is selected based on predefined probabilities.

3. **Group Population**:

- *For Binary Datasets*: An assay is sampled uniformly at random. A group is formed by sampling 1 binder and 4 decoys (1:4 ratio) from that assay.
- *For Affinity Datasets*: Assays are sampled with probability proportional to the Interquartile Range (IQR) of their affinity values. A group is formed by uniformly sampling $N$ ligands from the selected assay.

4. **Replacement**: Sampling is performed without replacement by default, automatically switching to replacement if the pool size is insufficient.

This strategy ensures that the model learns from consistent experimental contexts while balancing class distribution for screening tasks and prioritizing informative dynamic ranges for regression tasks.

## B.5  Optimizer Strategy

The models were trained with different optimization strategies to maximize performance on their respective tasks.

For the virtual screening task, we employed a hybrid optimization strategy by mixing two different optimizers. Specifically, for the parameters within the EGNN hidden layers, we used the **Muon** optimizer (Jordan et al., 2024). This allowed us to apply a high learning rate of $2 \times 10^{-3}$ and a weight decay of 0.01 to accelerate the convergence of the model's core representation learning component. All other model parameters (e.g., embedding layers, prediction head) were optimized using a standard **AdamW** optimizer (Kingma & Ba, 2017) with more conservative hyperparameters ($\beta_1 = 0.9$, $\beta_2 = 0.95$, and a weight decay of $1 \times 10^{-3}$).

However, for the more sensitive affinity value regression task, this aggressive, hybrid strategy led to training instability. We therefore opted for a simpler and more stable approach, using only the standard **AdamW** optimizer for all model parameters.

Finally, for the enzyme-substrate interaction task, we strictly adhered to the hyperparameter settings of the baseline method, EZSpecificity, to ensure a rigorous and fair comparison. Consequently, we employed the standard **AdamW** optimizer with a learning rate of $3 \times 10^{-4}$ for all model parameters.

## C   Experimental Protocols

This section provides the full experimental conditions for the prospective wet-lab validation, covering the *E. coli* DnaG inhibition assay, the whole-cell antibacterial assay, and compound sourcing.

### C.1   *E. coli* DnaG Inhibition Assay

Inhibition of *E. coli* DNA primase (dnaG) was assessed using an *in vitro* assay developed by ProFoldin (Hudson, MA), following the manufacturer's instructions. The assay is based on the measurement of the RNA primers synthesized by DNA primase in the presence of DNA template and NTPs. For screening experiments, reactions were performed using $40\mu l$ of reaction mixture including $24\mu l$ ultrapure Milli-Q water, $4\mu l$ of 10x assay buffer, $4\mu l$ of 10x DNA template, $4\mu l$ of 10x enzyme, and $4\mu l$ of 10x NTP mix, resulting in final concentrations of 10mM HEPES (pH 7.5), 5mM magnesium sulfate, 0.5mM dithiothreitol, 0.003% Brij-35, 100nM DNA, 0.5mM NTPs, and 100nM enzyme. $36\mu l$ of diluted buffer containing enzyme and NTP mix was plated into standard black 384-well plates (Corning 3575). Where applicable, $0.8\mu l$ of test compound (or DMSO as a negative control) was added, and plates were incubated at room temperature for at least 5min. Four $\mu l$ of 10x DNA template was then added to each reaction. For generating standard curves, the amount of substrate (DNA template) added was decreased in proportion to activity. Plates were incubated at 37°C for 2h. The provided $10\times$ fluorescence dye was diluted 10-fold with ultrapure Milli-Q water. After incubation, $60\mu l$ of 1 dye was added to each reaction, and mixtures were incubated at room temperature for 5min. The fluorescence excitation/emission at 485/535nm was then measured using a SpectraMax M3 plate reader. For each sample, activity was determined by linear interpolation with respect to the standard curves provided that the resulting fluorescence intensity value fell within the standard curve range. Otherwise,

Table 3: **Per-step inference-cost breakdown for FlashBind.** Per-target costs are amortized over a representative library of 5000 ligands per target. All timings are measured in-house on a single NVIDIA L40S GPU.

| Step | Cost type | Time |
|---|---|---|
| Protein structure retrieval/prediction | Per target (amortized) | 8.2 s/target $\rightarrow$ $1.6 \times 10^{-3}$ s/complex |
| FABind+ docking | Per complex | 0.67 s |
| Feature extraction (protein, ESM-3) | Per target (amortized) | 0.2 s/target $\rightarrow$ negligible |
| Feature extraction (ligand, torchdrug) | Per complex | negligible (>800 ligands/s per CPU) |
| Pocket cropping + graph construction + EGNN scoring | Per complex | $\sim$0.025 s |
| **End-to-end** | — | $\sim$0.70 s per complex |

fluorescence intensity values below that of the zero standard were clipped to that of the zero standard, and fluorescence intensity values above that of the highest standard were linearly extrapolated with respect to that of the highest standard. For subsequent validation dose–response experiments, half the indicated reaction volumes—that is, $20\mu$l for each reaction mixture—was used, and $40\mu$l of 1x dye was added to each reaction.

### C.2  *E. coli* Inhibition Assay

To test the antibacterial activity of each compound, we grow *E. coli* cells overnight in 3 mL LB medium and diluted 1/10000 into fresh LB. In 96-well flat-bottom plates (Corning), cells are then introduced to compound at an initial concentration of $128\mu$g/mL, either mixed or not mixed with $32\mu$g/mL polymyxin B nonapeptide. The plates are then incubated at 37°C without shaking until untreated control cultures reach stationary phase, at which time plates were read at 600 nm using a SpectraMax M3 plate reader. Cell viability values are normalized by the mean of two DMSO controls.

### C.3  Compound Preparation

Compounds with high purity were procured from the Broad Institute Center for the Development of Therapeutics.

## D  Efficiency Analysis

This section provides a fine-grained breakdown of FlashBind's inference cost, complementing the three-stage summary in Fig. 3d. All timings are measured on a single NVIDIA L40S GPU. We distinguish *per-complex* costs, incurred for every protein-ligand pair, from *per-target* costs, incurred once per protein target and amortized across the ligand library screened against it. Consistent with the standard high-throughput screening setting, in which a single target is screened against a large library, we amortize all per-target costs over a representative ratio of one target to 5000 ligands; Table 3 summarizes the resulting per-step costs.

**Protein structure retrieval/prediction (per target).** We measured this stage on 100 sequences sampled from the training set. Each target's structure is first sought through a retrieval cascade over the PDB and AlphaFold DB and, only as a last resort, predicted de novo with Boltz-2x. Retrieval verifies that a candidate sequence matches the target and that backbone coordinates are complete, then renumbers and aligns the accepted structure to the query; this verification and post-processing dominate the cost, while the initial entry search is comparatively cheap. A successful retrieval takes $\sim$5 s per sequence, whereas de novo Boltz-2x prediction takes $\sim$20 s. Of the 100 sequences, 78 were resolved through retrieval and only 22 required Boltz-2x, giving a weighted average of $\sim$8.2 s to obtain one structure. Incurred once per target and amortized over 5000 ligands, this contributes only $\sim$$1.6 \times 10^{-3}$ s per complex. This $\sim$20 s figure reflects Boltz-2x structure prediction alone, and is distinct from the $\sim$35 s reported for the Boltz-2 baseline, which corresponds to its full

Table 4: Inference time comparison per protein-ligand pair on a single NVIDIA L40S GPU. Our full pipeline offers a 50x speedup over Boltz-2. All timings are measured in-house under identical hardware and conditions.

| Method | Boltz-2 | BACPI | GAT | Chemgauss4 | AutoDock Vina | GNINA | Boltzina | FlashBind (Full Pipeline) | FlashBind (Scoring Only) |
|---|---|---|---|---|---|---|---|---|---|
| **Avg. Time (s)** | 35 | 0.00065 | 0.00028 | 25 | 4.5 | 5 | 6.5 | 0.7 | 0.025 |

affinity-prediction pipeline (structure prediction followed by a pocket re-pass through the trunk and affinity module).

**FABind+ docking (per complex).** Measured over 100 randomly sampled complexes, FABind+ docking takes ∼0.67 s per complex. This is the dominant per-complex component and, as noted in the main text, the sole remaining bottleneck.

**Feature extraction (per target and per complex).** Protein ESM-3 embeddings are computed once per target (∼0.2 s with batching) and are negligible after amortization. Ligand torchdrug features are computed per complex but are extremely lightweight, exceeding 800 ligands per second on a single CPU and fully parallelizable across workers during preprocessing.

**Pocket cropping, graph construction, and EGNN scoring (per complex).** The first two are rule-based data-loading operations that run almost for free when parallelized across data-loading workers; combined with the final EGNN scoring, these three steps together account for ∼0.025 s per complex.

Summing the per-complex components (FABind+ docking, ligand featurization, and cropping/graph/scoring) together with the negligible amortized per-target terms yields an end-to-end latency of ∼0.70 s per complex, consistent with the value reported in the main text.

**Library-scale projection.** These per-complex costs translate directly to library scale. Screening a one-million-compound library against a single target therefore requires approximately $0.70\,\text{s} \times 10^6 \approx 8.1$ GPU-days for the full pipeline, or $0.025\,\text{s} \times 10^6 \approx 7$ GPU-hours in the common rescoring setting where a pre-docked pose library already exists; the same library would require approximately $35\,\text{s} \times 10^6 \approx 405$ GPU-days for Boltz-2. At this scale the per-target one-time costs (structure acquisition and ESM-3 embedding, together under 10 s) are entirely negligible, confirming that the precomputation is amortized away.

**Cross-method comparison.** For completeness, Table 4 reports the average per-complex inference time of FlashBind against all baselines, measured on the same hardware and averaged over 100 randomly selected samples to account for variance in protein size and graph complexity.

# E Data Construction Analysis

This section reports two analyses of our virtual-screening training corpus: a comprehensive assessment of potential train/test leakage (Appendix E.1), and an ablation quantifying the contribution of the secondary-confirmation curation step (Appendix E.2).

## E.1 Data Leakage Analysis

To rigorously establish that FlashBind's performance is not an artifact of train/test contamination, we extend the protein- and ligand-level controls of Section 3.3 with a more comprehensive analysis spanning assay/dataset overlap, scaffold-series similarity, and target-family relatedness.

**Assay-level and PubChem/MF-PCBA overlap.** We verified that there is no intersection between the PubChem assay identifiers (AIDs) in our training corpus and those constituting the MF-PCBA test set. The two sets are disjoint at the assay level, so no test assay is seen during training. For the antibiotic benchmark (Wong et al., 2022), whose targets are not associated with PubChem AIDs, we separately confirmed

that none of its target proteins appears in our training set. Together these checks rule out assay-level and dataset-level leakage for both benchmarks.

**Scaffold-series leakage.** Our ligand-level control already discards any training ligand whose Tanimoto similarity (Butina, 1999) to a test active exceeds 0.4. This threshold follows established practice. The Lo-Hi Hit-Identification benchmark (Steshin, 2023) constructs its splits so that no test molecule exceeds ECFP4 (Rogers & Hahn, 2010) Tanimoto 0.4 to the training set, and RoseTTAFold All-Atom (Krishna et al., 2024) similarly treats ligands below 0.5 Tanimoto similarity as dissimilar to training. To quantify any residual overlap beyond this filter, we additionally computed Bemis–Murcko (Bemis & Murcko, 1996) scaffold overlap between test actives and the training set. Here, 26.2% (287/1094) of test actives share a molecular framework and 24.5% (229/935) share a deduplicated scaffold with some training compound. Bemis–Murcko scaffolds are, however, coarse. Privileged ring systems such as benzene, pyridine, and indole are ubiquitous across essentially all bioactive libraries, so some baseline overlap is unavoidable between any two unrelated activity sets. By construction, every scaffold-sharing pair already has pairwise Tanimoto below 0.4 and is well separated in fingerprint space, so the residual overlap reflects the prevalence of common ring systems rather than memorization of test actives. Our protocol is, if anything, more conservative than the foundation model we compare against. Boltz-2 (Passaro et al., 2025) applies no ligand-level similarity filtering (its authors report that such filtering does not affect their results), whereas we enforce a strict 0.4 Tanimoto cutoff, so any residual similarity-based leakage would tend to favor the baseline rather than FlashBind.

**Target-family-level leakage.** At the protein level, we remove any training protein sharing $\geq 90\%$ sequence identity with a validation or test protein, matching the deduplication protocol of Boltz-2. Because the benchmarks provide no explicit family labels, we further probed family-level relatedness by re-clustering all targets with MMseqs2 (Steinegger & Söding, 2017) at a much looser 30% sequence-identity threshold, well into the "twilight zone" of remote homology, where global fold may be shared but binding-site composition and geometry typically diverge. Even at this stringent threshold, only 1 of the 10 MF-PCBA test targets falls into a cluster that also contains a training target. As a direct check on its influence, we recomputed enrichment with this single target excluded and found the results essentially unchanged (EF@0.5%/1%/2%/5% $= 16.51/13.96/9.44/6.78$, vs. $16.28/14.13/9.46/6.73$ originally). We therefore retain it in the main results. Our leakage-control protocol is deliberately matched to that of Boltz-2, and removing remote homologs that the baseline does not would compromise the equivalence of evaluation conditions while penalizing the very cross-family generalization that virtual screening aims to achieve.

Taken together, these analyses at the assay, scaffold, and target-family levels indicate that our reported performance is not driven by data leakage.

## E.2  Secondary-Confirmation Ablation

Our PubChem curation pipeline (Section 3.3) applies three filtering stages, each targeting a distinct noise source: an assay-level filter, a compound-level secondary-confirmation step, and PAINS removal followed by class balancing, subsampling, and leakage prevention. Among these, secondary confirmation is the one most directly responsible for suppressing the false-positive active labels endemic to high-throughput screening. It retains an active label only when the label is corroborated by quantitative evidence, and it discards unconfirmed conflicting entries. This design follows the label-quality protocol of Boltz-2 (Passaro et al., 2025), whose authors report that roughly 40% of compounds labeled active in high-throughput primary screens may be false positives when cross-referenced against confirmatory screens.

To isolate the contribution of this step, we reconstructed the training set over the same assays with secondary confirmation removed, taking active labels directly from the primary-screening readouts without requiring quantitative corroboration. Inactive labels are unaffected, and all downstream stages (PAINS removal, 1:9 balancing, subsampling, and leakage prevention) are applied identically. The validation and test sets are held fixed across both runs, so only the training labels differ; we then retrain with identical hyperparameters and use the same top-2 checkpoint ensembling, inference, and evaluation protocol.

As expected, removing the filter substantially enlarges the active set. Active pairs grow from 213595 to 358723, the mean number of actives per assay increases by about 320, and the median rises sharply from 41

Table 5: **Ablation of the secondary-confirmation curation step on the full MF-PCBA benchmark.** Removing the step admits unconfirmed (likely false-positive) active labels and degrades every enrichment metric, most severely at EF@1%. Both models are trained and evaluated in-house under an identical protocol, differing only in the training labels.

| Training data | AUROC ↑ | EF@0.5% ↑ | EF@1% ↑ | EF@2% ↑ | EF@5% ↑ |
|---|---|---|---|---|---|
| Full pipeline | 0.7826 | 16.28 | 14.13 | 9.46 | 6.73 |
| w/o secondary confirmation | 0.7657 | 13.25 | 9.58 | 7.44 | 5.44 |

to 214.5. The newly admitted actives, those lacking quantitative confirmation, make up about 40% of the enlarged active set, matching the false-positive fraction estimated by Boltz-2 and indicating that this added mass is enriched for false positives.

The effect on downstream performance is summarized in Table 5. Removing secondary confirmation degrades every enrichment metric, and the largest effect falls on early enrichment. EF@1% drops from 14.13 to 9.58 (a 32% relative reduction), the metric most relevant to hit prioritization, while global AUROC declines only modestly (0.7826 to 0.7657). This pattern, with degradation concentrated at the top of the ranked list rather than in global discrimination, is what contaminating false-positive labels would be expected to produce, and it is the regime that matters most for virtual screening. The result confirms that secondary confirmation removes a meaningful fraction of noisy labels, and it supports our broader conclusion that a larger corpus helps only when paired with careful noise suppression, so data quantity and quality control must be pursued together.

# F    Detailed Results

This section provides the detailed numerical data corresponding to the figures presented in the main text, including specific Enrichment Factor (EF) values for virtual screening, precise inference speed measurements, and fine-grained performance breakdowns for enzyme families.

## F.1    Evaluation Protocol

To ensure the transparency and fairness of our comparisons, we document both how input structures are obtained for each method and the provenance of every reported value. **Throughout all results tables, a dagger (†) denotes values reproduced from prior publications; all unmarked entries are computed in-house.** In every case, reproduced and in-house results are evaluated on a common test set under an identical metric definition.

**Input structures.**    Because the compared methods are architecturally heterogeneous, the appropriate notion of a "shared input" differs by method type (Table 6). FlashBind and the traditional docking baselines receive identical protein inputs and differ only in their native docking engine, as each scoring function is designed to operate on the poses produced by that engine. Boltz-2, by contrast, is itself a structure-prediction model whose affinity head is trained on its own jointly predicted complexes. Imposing a single external structure on every method would therefore diminish rather than improve fairness, placing each scorer outside the structural regime for which it was designed; we accordingly evaluate each method within its native pipeline. We also isolate the effect of the structure source directly. The "Boltz-2 (FABind+)" variant in Table 9 supplies FlashBind's FABind+ poses to Boltz-2's scorer while holding the input fixed, and its performance remains close to that of native Boltz-2, confirming that our conclusions are not an artifact of differing structure-generation pipelines.

**Provenance of reported values.**    Table 7 summarizes the provenance of every reported result. Re-evaluating foundation models across the full MF-PCBA library is computationally prohibitive; we therefore adopt the baseline values reported in the original Boltz-2 publication for the full benchmark (Table 8), which additionally precludes any risk of under-tuning the baselines. The ablation and traditional-docking

Table 6: **Source of protein and complex structures for each in-house method.** Every scorer is evaluated within its native structural pipeline (AFDB: AlphaFold Database).

| Method | Protein structure | Complex structure |
|---|---|---|
| FlashBind (+ variants) | Retrieval or prediction (PDB / AFDB / Boltz-2x) | FABind+ docking |
| Traditional docking | Retrieval or prediction (PDB / AFDB / Boltz-2x) | AutoDock Vina docking |
| Boltz-2 (+ variants) | Jointly predicted with the ligand (end-to-end) | Jointly predicted (end-to-end) |

Table 7: **Provenance of reported results.** For each experiment, the methods computed in-house versus reproduced from prior work; in all cases both are evaluated on a common test set under an identical metric.

| Experiment | In-house | Reproduced from prior work |
|---|---|---|
| Full MF-PCBA (Fig. 3a, Table 8) | FlashBind | Boltz-2 and all remaining baselines (Boltz-2 paper) |
| Ablation & traditional baselines (Fig. 3b–c, Table 9) | All variants and traditional docking baselines | — |
| Enzyme-substrate (Fig. 4, Table 11) | FlashBind, Boltz-2 | EZSpecificity, ESP |
| Antibiotic discovery (Fig. 5) | FlashBind, Boltz-2 | AutoDock Vina and ML rescoring baselines |

Table 8: Performance comparison on the MF-PCBA benchmark for binder classification. Our model demonstrates competitive performance, particularly in enrichment factors, compared to established baselines. AUROC is computed globally across all targets. [†] Values reproduced from the original Boltz-2 publication; the unmarked FlashBind row is evaluated in-house on the identical test set and metric (Appendix F.1).

| Method | AUROC ↑ | EF at 0.5% ↑ | EF at 1% ↑ | EF at 2% ↑ | EF at 5% ↑ |
|---|---|---|---|---|---|
| Boltz-2[†] | 0.8056 | 18.3916 | 13.9540 | 10.5706 | 7.0448 |
| BACPI[†] | 0.7205 | 9.4818 | 9.2397 | 7.3983 | 5.5533 |
| GAT[†] | 0.7867 | 11.1279 | 8.9897 | 7.5630 | 5.9055 |
| Chemgauss4[†] | 0.5706 | 1.9969 | 2.2257 | 2.1136 | 1.6462 |
| Boltz-2 iptm[†] | 0.6134 | 2.4242 | 3.1728 | 2.6881 | 2.2263 |
| FlashBind | 0.7826 | 16.2832 | 14.1343 | 9.4587 | 6.7300 |

comparisons (Table 9) are instead computed entirely in-house on a common 50000-compound subset, yielding a strictly controlled head-to-head evaluation; as the enrichment factor is pool-dependent, its absolute magnitude on this subset differs from that on the full benchmark. For enzyme-substrate specificity and antibiotic discovery, both FlashBind and Boltz-2 are evaluated in-house on the exact benchmark test sets and metrics, whereas the remaining baselines are quoted from their original publications under each benchmark's standard protocol.

### F.2 Virtual Screening Performance

We present the comprehensive metrics for the MF-PCBA benchmark (Fig. 3 in main text). Table 8 details the performance of FlashBind against primary baselines, and Table 9 provides the detailed results for the ablation study and comparison against various docking and rescoring strategies.

Table 9: **Ablation study and extended comparison on MF-PCBA subset.** This table details the performance of various architectural variants and traditional docking methods. "Boltz-2 (FABind+)" indicates Boltz-2 scoring using FABind+ poses. "FlashBind (Boltz trunk)" uses the Boltz-2 trunk for feature extraction. AUROC is computed globally across all targets. All methods are evaluated in-house on a common 50000-compound subset under identical inputs and metric, with no values reproduced from external sources; as the enrichment factor is pool-dependent, absolute magnitudes differ from the full benchmark (Table 8).

| Method | AUROC ↑ | EF at 0.5% ↑ | EF at 1% ↑ | EF at 2% ↑ | EF at 5% ↑ |
|---|---|---|---|---|---|
| Boltz-2 | 0.7788 | 15.3087 | 10.4519 | 8.2599 | 7.1524 |
| Boltz-2 (FABind+) | 0.7852 | 16.1420 | 11.1186 | 7.7377 | 6.4675 |
| Boltzina | 0.7699 | 10.1308 | 10.0894 | 7.8458 | 6.9278 |
| AutoDock Vina | 0.5956 | 5.0000 | 3.2143 | 1.6071 | 2.8661 |
| AutoDock Vina (RF-Score) | 0.4899 | 0.0000 | 2.6800 | 2.2581 | 2.1325 |
| AutoDock Vina (RF-Score-VS) | 0.4668 | 0.0000 | 0.9271 | 0.4589 | 0.6463 |
| AutoDock Vina (PLEC Score) | 0.4817 | 0.0000 | 0.0000 | 0.0000 | 0.3568 |
| AutoDock Vina (NNScore) | 0.5062 | 0.0000 | 0.0000 | 0.3571 | 0.7389 |
| GNINA | 0.5592 | 2.7767 | 2.7543 | 4.0957 | 2.5896 |
| FlashBind (Boltz trunk) | 0.7553 | 16.0468 | 12.0458 | 7.5255 | 5.6767 |
| FlashBind | 0.7778 | 14.9905 | 12.8531 | 7.9154 | 6.7143 |

Table 10: **Per-target dispersion of enrichment on the MF-PCBA subset.** Mean ± standard deviation of the per-target enrichment factor across the 10 targets of the ablation subset (Table 9), for FlashBind and Boltz-2 (both evaluated in-house). The large standard deviations reflect genuine heterogeneity in target difficulty ($n = 10$) rather than model instability.

| Metric | FlashBind | Boltz-2 |
|---|---|---|
| EF@0.5% | $14.99 \pm 18.09$ | $15.31 \pm 20.45$ |
| EF@1% | $12.85 \pm 14.38$ | $10.45 \pm 10.80$ |
| EF@2% | $7.92 \pm 7.43$ | $8.26 \pm 7.05$ |
| EF@5% | $6.71 \pm 5.08$ | $7.15 \pm 4.77$ |

## F.3 Statistical Reproducibility and Dispersion

We first assess run-to-run reproducibility. Holding all inputs and parameters fixed, we re-ran the scoring pipeline under five distinct random seeds; every reported metric (global AUROC and EF@0.5/1/2/5%) was identical across the five runs, with raw prediction scores differing only at the fifth-to-sixth decimal place. FlashBind's reported results are therefore deterministically reproducible. Given the same checkpoint and inputs, inference returns the same outcome.

To characterize statistical dispersion, Table 10 reports the mean and standard deviation of the per-target enrichment factor across the 10 targets of the MF-PCBA ablation subset (Table 9), for both FlashBind and Boltz-2. FlashBind's dispersion is comparable to that of Boltz-2 at every threshold. The standard deviations are large relative to the means; this arises from the small number of targets ($n = 10$) and genuine heterogeneity in their difficulty (some targets are intrinsically easy and score highly for both methods, while others are hard for both), not from instability of any single model. The closely matched spread of the two methods supports this interpretation. We report the dispersion on this subset because it is the setting in which both methods were re-run in-house, providing per-target values for each (Appendix F.1).

## F.4 Enzyme Specificity Results

Table 11 provides the numerical breakdown of the enzyme-substrate specificity performance (AUROC) across five distinct enzyme families, corresponding to the radar plot in Fig. 4(b) of the main text. FlashBind demonstrates robust generalization, particularly in the DUF and Esterase families, outperforming the foundation model Boltz-2.

Table 11: **AUROC performance on specific enzyme families (ESIBank).** FlashBind consistently matches or outperforms baselines across diverse enzyme categories, including those with sparse data (e.g., DUF). [†] Values reproduced from prior work; unmarked rows (FlashBind and Boltz-2) are evaluated in-house on the identical benchmark split and metric (Appendix F.1).

| Method | Enzyme Family (AUROC) | | | | |
|---|---|---|---|---|---|
| | **DUF** | **Glycosyltransferase** | **Thiolase** | **Phosphatase** | **Esterase** |
| Boltz-2 (Binary Prob.) | 0.5468 | 0.7440 | 0.5307 | 0.6219 | 0.5755 |
| Boltz-2 (Affinity Val.) | 0.4409 | 0.6042 | 0.5545 | 0.4882 | 0.4101 |
| ESP[†] | 0.4594 | 0.5878 | 0.6439 | 0.5921 | 0.5091 |
| EZSpecificity[†] | 0.7067 | **0.7972** | 0.5970 | 0.6741 | **0.8035** |
| FlashBind | **0.7216** | 0.7688 | **0.7067** | **0.7010** | 0.7869 |

