# OpenReview forum: "FlashBind: Towards Accurate and Efficient Structure-based Virtual Screening"
_TMLR — Under review for TMLR_

### Review · Reviewer_Bvfs · 2026-06-17

**Summary Of Contributions:**

This paper proposes FlashBind, a lightweight structure-based virtual screening framework. The method replaces the expensive Boltz-2-style structure generation process with fast docking using FABind+, and replaces the heavier PairFormer module with an EGNN geometric scorer. The paper evaluates the method on MF-PCBA, ESIBank enzyme-substrate specificity prediction, an antibiotic discovery benchmark, and a prospective wet-lab DnaG screening campaign.
The problem setting of this paper has substantial practical value. It replaces the structure generation/scoring pipeline of heavy models such as Boltz-2 with FABind+ docking plus an EGNN scorer, with the goal of improving the speed of structure-based virtual screening. The overall idea is clear, and the systems engineering is relatively complete. In addition, the experiments cover MF-PCBA, ESIBank, an antibiotic benchmark, and DnaG wet-lab validation; the experimental setup is comprehensive and the application narrative is strong. In particular, the prospective wet-lab validation includes experimental measurements for 136 compounds and identifies 10 DnaG hits, which is more convincing than many papers that rely only on benchmark results.
The main weakness is that the methodological novelty is relatively limited. FlashBind is closer to a reasonable engineering combination: FABind+ generates poses, ESM-3/torchdrug provide features, and EGNN performs scoring. The individual modules themselves are not new, and the main contribution appears to come from system integration and data curation. In addition, some experimental results are not sufficiently transparent or rigorous, and the paper lacks adequate statistical evidence and experimental controls.

**Additional Comments:**

Overall, I find the application direction of this paper important, and the experimental scope has potential. However, I do not recommend accepting the paper in its current form. If the authors can address the critical changes above, this work has the potential to become a strong applied machine learning/drug discovery paper.

**Audience:**

Yes

**Audience Explanation:**

This paper addresses a problem that is clearly of interest to part of the TMLR audience: how to use geometric deep learning for efficient protein-ligand virtual screening. The system design that combines fast docking with a lightweight equivariant geometric scorer has practical significance, and the prospective experimental validation will also be of interest to researchers working at the intersection of machine learning, computational chemistry, and drug discovery.

**Broader Impact Concerns:**

This work concerns drug discovery and antibiotic screening. Its positive impact is that it may accelerate the discovery of potential antibacterial compounds. Potential risks include dual-use concerns associated with improved bioactivity screening models, overinterpretation of computational screening hits as validated therapeutic candidates, and possible misuse of model outputs in the absence of experimental confirmation. The paper should include a clearer broader impact statement discussing responsible use, the limitations of virtual screening, and the necessity of experimental validation.

**Claims And Evidence:**

No

**Claims Explanation:**

The paper provides some promising evidence, but I do not think the current evidence is sufficient to support the main claims made in the paper. One core claim is that FlashBind achieves accuracy comparable to Boltz-2. However, apart from EF@1% on MF-PCBA, where FlashBind reaches 14.13 and Boltz-2 reaches 13.95, FlashBind performs worse than Boltz-2 on EF@0.5%, EF@2%, and EF@5%, with a particularly noticeable gap at EF@0.5%. In addition, some baseline results are taken from prior publications, whereas FlashBind is trained and evaluated by the authors. The paper does not sufficiently demonstrate that there is no assay-level, target-family-level, or scaffold-level leakage between the PubChem-derived training data and MF-PCBA or the antibiotic benchmark.
Another core claim is that FlashBind is substantially faster than Boltz-2 at inference time: the paper states that Boltz-2 requires about 35 seconds per protein-ligand complex, while FlashBind requires only about 0.7 seconds. However, the current manuscript does not define the boundary of this timing measurement clearly enough. The paper appears to start FlashBind timing from the FABind+ docking stage, but before that stage there is also a Protein Structure Retrieval/Prediction step. The paper does not report in detail the time required by the three structure acquisition paths: PDB retrieval, AlphaFold DB structure selection, and Boltz-2x de novo prediction. Therefore, the speed comparison against Boltz-2 diffusion-based structure generation is not sufficiently transparent, and it is difficult to determine whether the claimed 50x speedup corresponds to a fair end-to-end inference pipeline.

**Requested Changes:**

Critical changes required for acceptance:
1. The authors should provide a stronger data leakage analysis. Beyond excluding leakage at the level of protein sequence similarity and active ligand similarity, the authors should also analyze possible assay-level, target-family-level, scaffold-series-level, and PubChem/MF-PCBA overlap.
2. The comparison with Boltz-2 and other baselines should be made more transparent and fair. The paper should clearly distinguish which baselines were reproduced by the authors and which numbers were taken from prior publications. For key baselines, the evaluation should, where possible, be conducted using the same input structures, the same compound set, and the same metric computation procedure.
3. The efficiency analysis should be explained more clearly. The authors should explicitly break down and report the time cost of each step, including Protein Structure Retrieval/Prediction, FABind+ docking, feature extraction, pocket cropping, graph construction, and EGNN scoring. They should also clarify which time costs are included in per-complex inference time and which costs are precomputed or amortized over each target.
Additional changes that would further strengthen the paper:
- Show failure cases, especially targets or chemotypes for which FlashBind performs poorly.
- It would be better to release the code, model weights, curated splits, and evaluation scripts.
- Weaken strong claims such as "high-fidelity screening" and "matches Boltz-2" unless stronger evidence is provided to support them.

---

> ### Author Response · Authors · 2026-06-23
> **Response to Reviewer Bvfs (1/4): Data Leakage Analysis**
>
> We appreciate your valuable feedback. Below are our detailed responses. We hope they address your concerns and welcome your favorable reconsideration.
>
> ---
>
> > ### **Data Leakage Analysis**
>
> We thank the reviewer for this constructive suggestion and have carried out a more comprehensive leakage analysis spanning assay/dataset overlap, scaffold-series similarity, and target-family relatedness, in addition to the protein- and ligand-level controls already in the paper. We summarize the results below.
>
> - **Assay-level and PubChem/MF-PCBA overlap.** We verified that there is **no intersection** between the PubChem assay identifiers (AIDs) in our training corpus and those constituting the MF-PCBA test set; the two are entirely disjoint at the assay level, so no test assay is seen during training. For the antibiotic benchmark, whose targets are not associated with PubChem AIDs, we separately confirmed that none of its target proteins appears in our training set. Together these checks rule out assay-level and dataset-level leakage for both benchmarks.
>
> - **Scaffold-series leakage.** Our ligand-level control already discards any training ligand whose Tanimoto similarity to a test active exceeds 0.4. This threshold follows established practice in the field: the Lo-Hi Hit-Identification benchmark (Steshin, NeurIPS 2023\) constructs its splits so that no test molecule exceeds ECFP4 Tanimoto 0.4 to the training set, and RoseTTAFold All-Atom (Krishna et al., 2024\) similarly treats ligands below 0.5 Tanimoto similarity as dissimilar to training. Our 0.4 cutoff therefore already removes the great majority of scaffold-similar compounds. To quantify any residual overlap beyond this filter, we additionally computed Bemis–Murcko scaffold overlap between test actives and the training set: 26.2% (287/1094) of test actives share a molecular structure and 24.5% (229/935) share a deduplicated scaffold. We note that Bemis–Murcko scaffolds are coarse: privileged ring systems such as benzene, pyridine, and indole are ubiquitous across essentially all bioactive libraries, so some baseline overlap is unavoidable between any two unrelated activity sets. Crucially, by construction every scaffold-sharing pair already has pairwise Tanimoto below 0.4 and is thus well separated in fingerprint space; the residual overlap therefore reflects the prevalence of common chemical scaffolds rather than memorization of test actives. We further note that our protocol is, if anything, more conservative than the foundation model we compare against: Boltz-2 applies no ligand-level similarity filtering (its authors report that such filtering does not affect their results), whereas we enforce a strict 0.4 Tanimoto cutoff, so any residual similarity-based leakage would tend to favor the baseline rather than FlashBind.
>
> - **Target-family-level leakage.** At the protein level, we remove any training protein sharing ≥90% sequence identity with a validation or test protein, matching the deduplication protocol used by Boltz-2 and standard practice in the field. Because the benchmarks provide no explicit family labels, we further probed family-level relatedness by re-clustering all targets at a much looser 30% sequence-identity threshold, well into the "twilight zone" of remote homology, where global fold may be shared but binding-site composition and geometry typically diverge. Even at this stringent threshold, only 1 of the 10 MF-PCBA test targets falls into a cluster also containing a training target. As a direct check on its influence, we recomputed enrichment with this single target excluded and found the results essentially unchanged (EF@0.5%/1%/2%/5% \= 16.51/13.96/9.44/6.78 vs. 16.28/14.13/9.46/6.73 originally); We therefore retain it in the main results: our leakage-control protocol is deliberately matched to that of Boltz-2, and removing remote homologs that the baseline does not would compromise the equivalence of evaluation conditions while penalizing the very cross-family generalization that virtual screening aims to achieve.
>
> Taken together, these analyses at the assay, scaffold, and target-family levels indicate that our results are not driven by data leakage. We thank the reviewer for prompting this deeper examination and will incorporate the full analysis into the revised manuscript.

---

> ### Author Response · Authors · 2026-06-23
> **Response to Reviewer Bvfs (2/4): Baseline Comparison**
>
> > ### **Baseline Comparison**
>
> We sincerely thank the reviewer for highlighting the importance of transparency and fairness in our baseline comparisons. We fully agree and clarify both how input structures are handled and the provenance of each reported number below.
>
> - **Input structures.** Because the compared methods are architecturally different, the appropriate notion of a "shared input" differs by method type. The table below summarizes how protein and complex structures are obtained for the results we ran.
>
> | Method | Protein structure | Complex structure |
> | :---- | :---- | :---- |
> | FlashBind (+ variants) | Retrieval/prediction (PDB / AFDB / Boltz-2x) | FABind+ docking |
> | Traditional docking | Retrieval/prediction (PDB / AFDB / Boltz-2x) | AutoDock Vina docking |
> | Boltz-2 (+ variants) | Jointly predicted with ligand (end-to-end) | Jointly predicted (end-to-end) |
>
> FlashBind and the traditional docking baselines therefore receive identical protein inputs; their complex structures follow each method's native docking engine, since each scoring function is designed to operate on poses produced by that engine. Boltz-2 is itself a structure-prediction model whose affinity head is trained on its own jointly predicted complexes. Forcing every method onto a single externally imposed structure would reduce rather than improve fairness, as it would place each scorer outside the structural regime it was built for; we therefore evaluate every method within its native pipeline.
>
> We did, however, directly isolate the effect of the structure source: the **"Boltz-2 (FABind+)"** variant in our ablation (Table 4\) feeds FlashBind's FABind+ poses into Boltz-2's scorer while holding the input fixed, and its performance remains close to native Boltz-2. This indicates that our conclusions are not an artifact of differing structure-generation pipelines.
>
> - **Provenance of reported numbers.** The following table specifies, for each experiment, which results we ran ourselves and which are cited from prior work. In all cases, self-run and cited methods are evaluated on the same test set with the same metric.
>
> | Experiment | Self-run by us | Cited from prior work |
> | :---- | :---- | :---- |
> | Full MF-PCBA (Fig. 3a, Table 3\) | FlashBind | Boltz-2 and all other baselines (original Boltz-2 paper) |
> | Ablation & traditional baselines (Fig. 3b–c, Table 4\) | All variants and traditional docking baselines | — |
> | Enzyme-substrate (Fig. 4, Table 6\) | FlashBind, Boltz-2 | EZSpecificity, ESP |
> | Antibiotic discovery (Fig. 5\) | FlashBind, Boltz-2 | AutoDock Vina and ML rescoring baselines |
>
> For completeness, we briefly explain the reasoning behind these choices in each setting:
>
> - **Full MF-PCBA:** re-evaluating foundation models over the entire library is computationally prohibitive, so we reference the baseline numbers from the original Boltz-2 publication. Using the authors' own reported numbers also avoids any risk of under-tuning the baselines on our end.
> - **Ablation & traditional baselines:** running these variants and traditional docking pipelines on the full 500k-compound library is prohibitively expensive, so we evaluate them on a representative 50k-compound subset (sampled per target proportionally to hit rate). Here every method is run by us on the same compound set with the same metric, giving a strictly controlled head-to-head comparison. As enrichment factor is pool-dependent, absolute values on the subset differ from those on the full benchmark.
> - **Enzyme-substrate & antibiotic discovery:** we locally ran both FlashBind and Boltz-2 on the identical test sets and metrics defined by each benchmark, while the remaining baselines are taken from their original publications under each benchmark's standard protocol, ensuring every competing method is represented by its authors' best-tuned configuration.
>
> We thank the reviewer again for raising this point, and in the revised manuscript we will annotate every results table to mark each number as self-run or cited and state the matched-evaluation conditions explicitly in each caption.

---

> ### Author Response · Authors · 2026-06-23
> **Response to Reviewer Bvfs (3/4): Efficiency Analysis**
>
> > ### **Efficiency Analysis**
>
> We thank the reviewer for this suggestion and agree that a clearer breakdown strengthens the efficiency analysis. We note that Figure 3d already reports per-stage wall-clock times for the three high-level stages (structure generation, feature extraction, scoring); below we provide a finer-grained breakdown of every individual step, together with how each cost is measured and whether it is incurred per complex or amortized over each target.
>
> All timings are measured on a single NVIDIA L40S GPU. The amortized column reflects the standard high-throughput virtual screening setting, in which a single protein target is screened against a large ligand library; we use a representative ratio of one target to 5,000 ligands.
>
> | Step | Cost type | Time |
> | :---- | :---- | :---- |
> | Protein structure retrieval/prediction | Per target (amortized) | 8.2 s per target → 0.00164 s per complex |
> | FABind+ docking | Per complex | 0.67 s |
> | Feature extraction (protein, ESM-3) | Per target (amortized) | 0.2 s per target → negligible per complex |
> | Feature extraction (ligand, torchdrug) | Per complex | negligible (\>800 ligands/s per CPU; parallelized) |
> | Pocket cropping \+ graph construction | Per complex | negligible (rule-based, parallelized across workers) |
> | EGNN scoring | Per complex | included in the 0.025 s below |
> | Inference *(Cropping \+ graph \+ scoring combined)* | Per complex | \~0.025 s |
> | **End-to-end per complex** | — | **\~0.70 s** |
>
> We elaborate on each step below.
>
> - **Protein structure retrieval/prediction (amortized per target).** We evaluated this stage on 100 sequences sampled from the training set. Each target's structure is first sought through a retrieval pipeline over the PDB and AlphaFold DB, and only as a last resort predicted de novo with Boltz-2x. Retrieval is treated as a single stage: it searches both sources for candidate entries and, for each candidate, verifies that the sequence matches the target and that backbone atom coordinates are not missing, then post-processes the accepted structures (renumbering and aligning residues to the query sequence) and selects the best one meeting all criteria. The verification and post-processing dominate the cost; the initial entry search is comparatively cheap. On a successful retrieval this takes ~5 s per sequence, while de novo Boltz-2x prediction takes ~20 s. Of the 100 sequences, 78 were resolved through retrieval and only 22 required Boltz-2x prediction. Weighting by this distribution gives an average of ~8.2 s to obtain one protein structure. Crucially, this cost is incurred once per target and amortized across the entire ligand library; at a ratio of one target to 5,000 ligands, the amortized per-complex cost is only ~0.00164 s and is therefore negligible.
>
>   We also clarify a related point on the Boltz-2 baseline timing. The \~20 s above reflects Boltz-2x structure prediction alone, whereas the \~35 s reported for Boltz-2 in the main text corresponds to its full affinity-prediction pipeline: after the initial structure is predicted, the pocket is cropped and re-passed through the full model together with the affinity module for scoring. The two numbers are therefore consistent and measure different quantities.
>
> - **FABind+ docking (per complex).** Measured over 100 randomly sampled complexes, FABind+ docking takes \~0.67 s per complex. This is the dominant component of the per-complex cost and, as noted in the paper, the sole remaining bottleneck.
>
> - **Feature extraction (amortized \+ per complex).** Protein ESM-3 embeddings are computed once per target and amortized: with batching, one protein's embedding takes \~0.2 s, which is negligible after amortization across the ligand library. Ligand TorchDrug features are computed per complex but are extremely lightweight, exceeding 800 ligands per second on a single CPU, and fully parallelizable across CPUs during preprocessing (negligible even at modest core counts). Together these account for the feature-extraction stage shown in Figure 3d, which we conservatively bound at under 2 ms per complex.
>
> - **Pocket cropping, graph construction, and EGNN scoring (per complex).** The first two are rule-based dataset-loading operations performed during inference and run almost for free when parallelized across data-loading workers. Combined with the final EGNN scoring, these three steps together account for \~0.025 s per complex.
>
> Summing the per-complex components (FABind+ docking, ligand featurization, and cropping/graph/scoring) together with the negligible amortized terms yields an end-to-end latency of \~0.70 s per complex, consistent with the value reported in the paper.
>
> We thank the reviewer again for prompting this clarification, and we will incorporate the full per-step breakdown and the amortization analysis above into the revised manuscript.

---

> ### Author Response · Authors · 2026-06-23
> **Response to Reviewer Bvfs (4/4): Other Questions**
>
> > ### **Failure Case**
>
> We thank the reviewer for this valuable suggestion, and we agree that an explicit analysis of failure cases strengthens the paper. We have examined how FlashBind's performance varies across targets, and we find that it tracks the definability of the binding site, supporting the limitation we hypothesized in our Discussion.
>
> This contrast is clearest on the antibiotic benchmark (Figure 5). The strongest targets are metabolic enzymes with well-defined small-molecule binding pockets, glmU (AUROC 0.90), murF (0.84), gmk and murC (0.80), whereas the weakest are large nucleic-acid–processing enzymes that act on extended interfaces and are difficult substrates for conventional docking: DNA gyrase (gyrA/gyrB, 0.46), DNA ligase (ligA, 0.48), and the replicative helicase (dnaB, 0.58). A consistent trend holds on MF-PCBA, where the weakest targets are proteins whose functional sites are shallow protein-interaction or regulatory surfaces rather than classical druggable pockets, such as a protein-CTD phosphatase (CTDSP1, AUROC 0.68) and an E2 ubiquitin-conjugating enzyme (UBE2N, 0.60).
>
> This tendency is consistent with the limitation we hypothesized in our Discussion: for targets that lack a clearly defined pocket, the upstream FABind+ stage is more likely to struggle in localizing the binding site and producing a reliable pose, and any resulting geometric error propagates to the downstream EGNN scorer. In other words, FlashBind's predictive ceiling is bounded by the fidelity of the upstream pose, as anticipated in our analysis.
>
> We note that these are correlational observations across targets rather than a controlled mechanistic attribution, since ground-truth binding poses are not available; a more complete mechanistic analysis would be a valuable direction for future work. We will incorporate this analysis into the revised manuscript.
>
> > ### **Open-source**
>
> We fully agree that releasing these materials strengthens reproducibility. We are committed to publicly releasing the complete pipeline, including the curated dataset splits, the model and training code, trained model weights, and evaluation scripts, so that all results in the paper can be reproduced end to end. We are currently preparing an anonymized snapshot of the repository and will aim to share a link during the discussion period if it is ready in time; in any case, the full materials will be made publicly available with the camera-ready release.
>
> > ### **"high-fidelity screening, match Boltz-2” Strong Claim**
>
> We appreciate the reviewer’s constructive feedback regarding the tone of our claims. We recognize that while FlashBind matches Boltz-2 at the critical 1% early enrichment threshold, there is a performance gap at other thresholds such as 0.5% and 2%. In our revised manuscript, we will systematically tone down strong phrases such as "matches Boltz-2" and "high-fidelity screening". Instead, we will use more precise language, framing our performance as achieving "competitive early enrichment" and explicitly discussing the trade-offs between screening speed and accuracy across different enrichment thresholds.
>
> > ### **Broader Impact**
>
> We thank the reviewer for highlighting the necessity of a comprehensive broader impact statement. We completely agree that the risks associated with AI in drug discovery must be clearly communicated. In our revision we will discuss the dual-use concerns associated with bioactivity screening models, including the possibility that a model ranking compounds by predicted target binding could in principle be redirected toward harmful ends such as the discovery of toxic agents, while clarifying that FlashBind is a scoring and ranking model operating over existing chemical libraries, not a generative tool that designs novel molecules or bioweapons de novo, so its misuse potential is bounded by the libraries it is applied to. We will further emphasize the inherent limitations of computational virtual screening: realizing any predicted activity still requires a resolved target structure, a synthesizable compound, chemical synthesis, and experimental confirmation, and model outputs should not be overinterpreted as validated therapeutic candidates without rigorous wet-lab experimental confirmation.
>
> ---
>
> We hope this response could answer your questions and address your concerns, looking forward to receive your further feedback soon.

---

### Review · Reviewer_AFNc · 2026-06-21

**Summary Of Contributions:**

The paper introduces FlashBind, a fast protein-ligand bind prediction algorithm. FlashBind successfully combines prior modules, namely (1) FABind+ for fast docking, (2) ESM-3, TorchDrug and RDKit for ligand and protein graph construction, (3) EGNN for E(3)-equivariant embedding and (4) a task-specific MLP that is trained on a curated dataset. This pipeline avoids the heavy computation of physics-based geometrical construction. The paper evaluates FlashBind on MF-PCBA benchmark, ESIBank benchmark, E. coli benchmark and a prospective campaign with a wet lab. Overall, FlashBind is shown to provide a drastic acceleration with limited performance reduction, which makes it promising for practical use on large-scale screening.

**Additional Comments:**

I am not an expert of this field.

**Audience:**

Yes

**Audience Explanation:**

The advancement seems significant enough to be interesting to TMLR's audience.

**Broader Impact Concerns:**

The paper does not discuss potential misuse of this technology, which seems to involve bioweapons.

**Claims And Evidence:**

Yes

**Claims Explanation:**

The paper provides a wide evaluation of FlashBind. I have two caveats though:
- The code does not seem to be provided, which hinders reproducibility.
- The authors also made a significant effort to construct a clean training dataset. It would have been interesting to see the extent to which their interventions on the dataset is significant for the performance of FlashBind. Typically, how would FlashBind have performed without the secondary confirmation?

**Requested Changes:**

Can the authors share datasets, algorithms and experimental setup? This seems important for reproducibility.

Could the authors provide an ablation study of the training dataset construction? While not essential, this would provide valuable insights into the importance of dataset construction.

---

> ### Author Response · Authors · 2026-06-29
> **Response to Reviewer AFNc (1/2): Data Construction**
>
> Thank you for acknowledging our work and the valuable feedbacks. In the following, we have carefully crafted detailed responses to address your comments.
>
> ---
>
> > ### **Data construction**
>
> We thank the reviewer for this suggestion. Quantifying the contribution of our curation steps is a valuable addition, and we have run the requested ablation on the most critical step, secondary confirmation.
>
> Raw PubChem BioAssays pass through three filtering stages, each targeting a distinct noise source: (i) an assay-level filter retaining only confirmatory/primary screens with more than 100 compounds and a hit rate below 10%, with same-target assays merged by UniProt ID to maximize chemical diversity; (ii) a compound-level secondary-confirmation step that keeps an active label only if it is corroborated by quantitative evidence $(K_d/K_i/IC_{50})$, with unconfirmed conflicting entries discarded as a secondary effect; and (iii) PAINS removal, followed by 1:9 balancing, subsampling and leakage prevention. Among these, secondary confirmation is the step most directly responsible for suppressing the false-positive active labels endemic to high-throughput screening, and this is its dominant purpose. Our design is referenced from the label-quality protocol of Boltz-2, which reports that, by cross-referencing confirmatory (secondary) screens, roughly 40% of compounds labeled active in high-throughput primary screens may be false positives, motivating an analogous confirmation step in our pipeline.
>
> To isolate this step, we reconstructed the training set over the same assays with secondary confirmation removed: active labels are taken directly from the primary screening readouts without requiring quantitative $K_d/K_i/IC _{50}$ corroboration. Inactive labels are unaffected by this step and left unchanged, and all downstream stages (PAINS removal, balancing, leakage prevention, subsampling) are applied identically. The validation and test sets are held fixed across both runs, so only the training labels differ. We then retrain with identical hyperparameters and use the same top-2 checkpoint ensembling, inference, and evaluation protocol.
>
> As expected, removing the filter substantially enlarges the active set: active pairs grow from 213595 to 358723, the mean number of actives per assay increases by approximately 320, and the median rises sharply from 41 to 214.5. Notably, the newly admitted actives, precisely those lacking quantitative confirmation, constitute about 40% of the enlarged active set, matching the false-positive fraction estimated by Boltz-2 and indicating that this added mass is enriched for false positives.
>
> On the full MF-PCBA benchmark, removing secondary confirmation degrades every enrichment metric, with the largest effect on early enrichment:
>
> | Training data              | AUROC  | EF@0.5% | EF@1% | EF@2% | EF@5% |
> | :------------------------- | :----- | :------ | :---- | :---- | :---- |
> | Full pipeline              | 0.7826 | 16.28   | 14.13 | 9.46  | 6.73  |
> | w/o secondary confirmation | 0.7657 | 13.25   | 9.58  | 7.44  | 5.44  |
>
> The drop is most pronounced at EF@1% (14.13 → 9.58, a 32% relative reduction), the metric relevant to hit prioritization, while global AUROC declines only modestly (0.7826 → 0.7657). This pattern is consistent with false positives degrading top-of-list ranking specifically, rather than global discrimination, which is exactly the regime that matters for virtual screening. The result confirms that secondary confirmation removes a meaningful fraction of contaminating labels and directly supports our Discussion: an expanded corpus is effective only when paired with rigorous noise suppression, data quantity and quality control must be pursued together. We will add this ablation to the appendix and reference it from 3.3.

---

> ### Author Response · Authors · 2026-06-29
> **Response to Reviewer AFNc (2/2): Other Questions**
>
> > ### **Open-source**
>
> We appreciate the reviewer's emphasis on reproducibility, which we fully share. We will publicly release the entire FlashBind pipeline, the curated dataset splits, the model and training code, trained checkpoints, and the evaluation scripts, so that every result in the paper can be reproduced end to end. An anonymized snapshot of the repository is being prepared, and we will post a link during the discussion period if it is finalized in time; in any case, the complete materials will accompany the camera-ready release under a public license. We also note that Appendices B and C already document the node featurization, pocket cropping, edge construction, group-based sampling, optimizer settings, and the full wet-lab protocols in detail.
>
> > ### **Broader Impact**
>
> We thank the reviewer for underscoring the importance of a thorough broader-impact discussion, and we agree that dual-use risks in AI-driven drug discovery deserve explicit treatment. Our revision will add such a statement. We will acknowledge that a model ranking compounds by predicted target binding could, in principle, be repurposed toward harmful objectives such as identifying toxic agents. At the same time, we will clarify that FlashBind is a scoring-and-ranking model operating over pre-existing chemical libraries, not a generative system that designs novel molecules de novo, so its misuse potential is bounded by the libraries to which it is applied. We will further stress the practical limits of computational screening: any predicted activity must still be realized through a resolved target structure, a synthesizable compound, actual chemical synthesis, and experimental confirmation, and model scores should not be mistaken for validated therapeutic candidates absent rigorous wet-lab verification.
>
> ---
>
> We hope this response could answer your questions and address your concerns, looking forward to receive your further feedback soon.

---

### Review · Reviewer_3ie3 · 2026-07-01

**Summary Of Contributions:**

This paper presents FlashBind, a lightweight geometric deep learning framework for structure-based virtual screening that achieves comparable accuracy to the foundation model Boltz-2 while being 50× faster at inference time (0.7s vs 35s per complex). The main contributions are the following:

1- A two-stage pipeline decoupling structure generation (using FABind+ for fast docking) from scoring (using an E(3)-equivariant GNN), eliminating expensive diffusion-based generation and PairFormer modules.

2- State-of-the-art enrichment on the MF-PCBA benchmark (EF@1% = 14.13), matching Boltz-2 (13.95) while substantially outperforming physics-based and sequence-based baselines.

3- Strong generalization to enzyme-substrate specificity prediction on ESIBank (AUROC 0.7229), on par with the specialized EZSpecificity model.

4- Prospective experimental validation: screening 9,289 compounds against E. coli DnaG, testing 136 compounds, and identifying 10 active inhibitors (7.4% hit rate), with 4 compounds showing whole-cell antibacterial activity.

5- Extensive ablation studies confirming that neither diffusion sampling nor heavy PairFormer architectures are necessary for high accuracy.

Strength points:

1- The 50× speedup is practically significant for industrial-scale virtual screening

2- Prospective wet-lab validation is a major strength, demonstrating real-world applicability

3- Thorough benchmarking across multiple tasks (virtual screening, enzyme specificity, antibiotic discovery)

4- Careful data curation pipeline to mitigate experimental noise

5- Clear ablation studies validating architectural choices

Weakness points:

1- The framework's performance is bounded by the upstream docking oracle (FABind+), failure cases at cryptic binding sites or with conformational plasticity would propagate errors.

2- While matching Boltz-2 on classification tasks, FlashBind does not fully match Boltz-2 on affinity regression (though ablation suggests this is due to training data provenance rather than architecture).

3- The method relies on pre-computed protein embeddings (ESM-3) which are amortized but require upfront computation.

**Audience:**

Yes

**Audience Explanation:**

This paper would be of significant interest to the TMLR audience for several reasons:

1- The paper addresses the fundamental efficiency-accuracy trade-off in machine learning, which is a core concern across the entire TMLR community. While the application is computational drug discovery, the architectural insights (lightweight equivariant networks can replace expensive transformers without performance loss) generalize to other domains.

2- The systematic approach to decoupling structure generation from scoring, the careful ablation studies, and the rigorous benchmarking provide a template for efficient model design that would interest ML researchers.

3- The prospective wet-lab validation demonstrates that ML methods can genuinely accelerate scientific discovery, which is a compelling narrative for applied ML.

4- This work sits at the intersection of geometric deep learning, efficiency optimization, and scientific applications, all areas of active interest to TMLR's diverse readership.

Open challenges: The paper candidly discusses limitations (bounded by docking oracle fidelity and affinity regression gap), which opens avenues for future work that would interest ML researchers.

**Broader Impact Concerns:**

No significant concerns identified.

**Claims And Evidence:**

Yes

**Claims Explanation:**

The authors provide empirical support for their claims:

1- Accuracy-efficiency trade-off: Fig. 1 and Fig. 3d clearly demonstrate the 50× speedup (0.7s vs 35s) with detailed per-component time breakdowns. The 50× claim is well-documented with actual wall-clock times.

2- Benchmark performance: Results on MF-PCBA (Table 3) show FlashBind achieving EF@1% = 14.13 vs Boltz-2's 13.95, with proper baseline comparisons including physics-based (Chemgauss4) and sequence-based (BACPI, GAT) methods.

3- Ablation studies: The step-wise ablations in Fig. 3b convincingly validate each design choice, confirming that FABind+ poses suffice (Boltz-2(FABind+) vs Boltz-2), EGNN matches heavier architectures (FlashBind vs FlashBind(Boltz trunk)), and lightweight embeddings work (FlashBind vs. FlashBind(Boltz trunk)).

4- Enzyme specificity: Results on ESIBank (Fig. 4a) show AUROC 0.7229 vs EZSpecificity 0.7198 and ESP 0.6523 with fine-grained family-level analysis.

5- Prospective validation: The wet-lab campaign is the strongest evidence, 10/136 confirmed actives (7.4% hit rate) with 4 showing whole-cell activity, demonstrating genuine translational utility.

6- Affinity regression: Appendix A provides thorough benchmarking on OpenFE, FEP+, and CASP16, with proper molecular-weight bias correction and architectural ablations.

**Requested Changes:**

1- Expand discussion of docking oracle failures: The paper acknowledges this limitation but could provide more detail. What is the failure rate of FABind+ on the benchmarks? Are there specific protein classes where it struggles?

2- The paper acknowledges the limitation of docking oracle failures but could provide more detail. What is the failure rate of FABind+ on the benchmarks? Are there specific protein classes where it struggles?

3- While per-complex time is reported, practical screening involves library-level operations. Please report the total time to screen 1M compounds, including protein embedding precomputation amortization.

4- Section 4.4 mentions PAINS filtering but doesn't detail how this was implemented. Please specify the PAINS substructure set used.

5- For key metrics (EF@1%, AUROC), please add standard deviations or confidence intervals across multiple runs/random seeds to demonstrate statistical significance.

6- Appendix A shows FlashBind underperforms Boltz-2 on affinity regression. While the authors attribute this to training data provenance, this remains a limitation. Please add a clear discussion of this limitation in the main text (perhaps in Section 5) rather than relegating it to the appendix.

7- The authors mention evaluating FlashBind "without finetuning" on the antibiotic discovery benchmark (Section 4.3), but it's unclear whether the model was trained on the MF-PCBA dataset and then directly applied or if there was any additional training. Please explicitly state the training scheme for each benchmark.

---

> ### Author Response · Authors · 2026-07-06
> **Response to Reviewer 3ie3 (1/3): Failure Case and Efficiency Analysis**
>
> Thank you for dedicating your time and providing valuable feedback. In the following, we have carefully crafted detailed responses to address your comments.
>
> ---
>
> > ### **Failure Case**
>
> We thank the reviewer for this valuable suggestion, and we agree that an explicit analysis of failure cases strengthens the paper. We first note that a strict docking failure rate cannot be measured on these benchmarks, as ground-truth binding pockets and poses are not available to verify whether FABind+ localizes the site correctly. Instead, we characterize failure through how FlashBind's performance varies across targets, and we find that **it tracks the definability of the binding site**, supporting the limitation we hypothesized in our Discussion.
>
> This contrast is clearest on the antibiotic benchmark (Figure 5). The strongest targets are metabolic enzymes with well-defined small-molecule binding pockets, glmU (AUROC 0.90), murF (0.84), gmk and murC (0.80), whereas the weakest are large nucleic-acid–processing enzymes that act on extended interfaces and are difficult substrates for conventional docking: DNA gyrase (gyrA/gyrB, 0.46), DNA ligase (ligA, 0.48), and the replicative helicase (dnaB, 0.58). A consistent trend holds on MF-PCBA, where the weakest targets are proteins whose functional sites are shallow protein-interaction or regulatory surfaces rather than classical druggable pockets, such as a protein-CTD phosphatase (CTDSP1, AUROC 0.68) and an E2 ubiquitin-conjugating enzyme (UBE2N, 0.60).
>
> This tendency is consistent with the limitation we hypothesized in our Discussion: for targets that lack a clearly defined pocket, the upstream FABind+ stage is more likely to struggle in localizing the binding site and producing a reliable pose, and any resulting geometric error propagates to the downstream EGNN scorer. In other words, FlashBind's predictive ceiling is bounded by the fidelity of the upstream pose, as anticipated in our analysis. The failure mode is therefore not uniformly distributed but concentrated in an identifiable protein class, targets acting on extended or poorly-defined interfaces, which is precisely the regime where pocket-based docking is least reliable.
>
> We note that these are correlational observations across targets rather than a controlled mechanistic attribution, since ground-truth binding poses are not available; a more complete mechanistic analysis would be a valuable direction for future work. We will incorporate this analysis into the revised manuscript.
>
> > ### **Efficiency Analysis**
>
> We thank the reviewer for this suggestion, and we agree that a library-level cost analysis better reflects practical screening usage than per-complex timing alone.
>
> We report the total cost of screening a **1M-compound library against a single target**, which is the standard high-throughput virtual screening setting. All timings are measured on a single NVIDIA L40S GPU.
>
> The cost decomposes into two parts. **Per-target one-time costs** (amortized over the entire library): **protein structure retrieval/prediction takes \~8.2 s per target** (averaged over our retrieval cascade of PDB → AlphaFold DB → Boltz-2x), and **ESM-3 protein embedding precomputation takes \~0.2 s per target**. These are incurred once and amortized across all 1M ligands, contributing under 10 s in total, i.e. entirely negligible at library scale. **Per-complex costs** are dominated by FABind+ docking (\~0.67 s), with ligand featurization, pocket cropping, graph construction, and EGNN scoring together adding only \~0.025 s.
>
> This yields the following library-level projection for 1M compounds:
>
> | Setting | Per-complex cost | Total for 1M compounds |
> | :---- | :---- | :---- |
> | **FlashBind (full pipeline)** | \~0.70 s | **\~8.1 GPU-days** |
> | **FlashBind (scoring only, pre-docked library)** | \~0.025 s | **\~7 GPU-hours** |
> | Boltz-2 (full pipeline) | \~35 s | \~405 GPU-days |
>
> Two points are worth highlighting. First, **the protein-embedding and structure-retrieval precomputation the reviewer asks about is genuinely amortized away**: spread over 1M ligands, its per-complex contribution is \~1.7×10⁻³ s, negligible relative to the \~0.7 s docking-dominated per-complex cost. Second, in the common **rescoring** scenario where a docked-pose library already exists, only the \~0.025 s scoring step is incurred, enabling **over 140,000 complexes per GPU-hour** and making million-scale campaigns tractable on modest hardware. Against Boltz-2's \~405 GPU-days for the same 1M-compound library, FlashBind's \~8.1 GPU-days (full pipeline) or \~7 GPU-hours (rescoring) is what makes the 50× per-complex speedup translate into a decisive practical advantage at library scale.
>
> A complete per-step breakdown of every individual component, together with how each cost is measured and the amortization analysis, is provided in our Response to Reviewer Bvfs (3/4): Efficiency Analysis. We will incorporate the library-level projection above into the revised manuscript.

---

> ### Author Response · Authors · 2026-07-06
> **Response to Reviewer 3ie3 (2/3): PAINS Filter, Statistical Confidence and Limitation of Affinity Regression**
>
> > ### **PAINS Filter**
>
> We thank the reviewer for pointing out this omission, and we agree that the PAINS filtering step should be specified explicitly.
>
> We use the standard **PAINS substructure set of Baell & Holloway (2010)**, as implemented in RDKit's `FilterCatalog` module. Concretely, we instantiate a `FilterCatalog` with `FilterCatalogParams.FilterCatalogs.PAINS`, which loads the **full PAINS catalog (the combined PAINS\_A, PAINS\_B, and PAINS\_C sub-catalogs, comprising 480 substructure filters)**. A compound is flagged as a PAINS hit if RDKit reports any match against this catalog, and flagged compounds are removed from the training corpus; the same filter is applied to exclude PAINS compounds from the prospective DnaG screening library (Section 4.4). This follows the PAINS-filtering protocol adopted by Boltz-2, ensuring consistency with the baseline.
>
> We will add these implementation details, including the exact RDKit catalog used, to the revised manuscript (Section 4.4 and Appendix C).
>
> > ### **Statistical Confidence**
>
> We thank the reviewer for this helpful suggestion, and we agree that quantifying the stability of our reported metrics strengthens the paper.
>
> We first assess run-to-run reproducibility. Keeping all other inputs and parameters unchanged, we re-ran the scoring pipeline under five different random seeds. All reported metrics (EF@0.5/1/2/5% and global AUROC) were identical across the five runs, and even the raw prediction scores differed only at the fifth-to-sixth decimal place or below. This confirms that FlashBind's reported results are deterministically reproducible: given the same parameters and inputs, inference yields the same outcome.
>
> To further characterize statistical dispersion, we computed the standard deviation of the per-target EF across the 10 targets of the **MF-PCBA subset (Table 4\)**, for both FlashBind and Boltz-2:
>
> | Metric | FlashBind | Boltz-2 |
> | :---- | :---- | :---- |
> | EF@0.5% | 14.9905 ± 18.0871 | 15.3087 ± 20.4541 |
> | EF@1% | 12.8531 ± 14.3765 | 10.4519 ± 10.8023 |
> | EF@2% | 7.9154 ± 7.4273 | 8.2599 ± 7.0512 |
> | EF@5% | 6.7143 ± 5.0772 | 7.1524 ± 4.7733 |
>
> The dispersion of FlashBind is comparable to that of Boltz-2 across all thresholds. We note that the standard deviations are large relative to the means; this reflects the small number of targets (n \= 10\) together with genuine heterogeneity in target difficulty, some targets are intrinsically easier and yield high EF for both methods, while others are hard for both, rather than instability of any single model. That both methods exhibit similar spread supports this interpretation: the variability arises from the targets themselves. The matched dispersion comparison is provided here on the subset because it is the setting where we re-ran Boltz-2 and thus have per-target values for both methods (see our Response to Reviewer Bvfs (2/4): Baseline Comparison for the full breakdown of which experiments we re-ran versus quoted from the literature).
>
> We will incorporate the seed-reproducibility statement and the above dispersion analysis into the revised manuscript.
>
> > ### **Limitation of Affinity Regression**
>
> We thank the reviewer for this suggestion, and we agree that this limitation belongs in the main text rather than only in the appendix. We will add an explicit discussion to Section 5 stating that, on absolute affinity regression, FlashBind does not consistently match Boltz-2 (Appendix A, Table 1). We will frame this candidly as a genuine limitation of the current model rather than a resolved issue.
>
> Regarding the source of the gap, we will be careful to present our attribution as a hypothesis rather than a definitive conclusion. Our encoder-architecture ablation (Appendix A.4) shows that replacing the EGNN with a heavier PairFormer yields negligible improvement, which lets us reasonably rule out limited architectural capacity as the primary cause. Based on this, we hypothesize that the difference is driven mainly by training-data provenance, in particular Boltz-2's larger and more rigorously curated proprietary affinity dataset. We will make clear, however, that this is an empirical inference and that other factors we did not fully control, such as differences in input preprocessing or labeling pipelines, may also contribute; we simply regard data provenance as the most likely dominant factor. We will incorporate this discussion into Section 5 of the revised manuscript, with a pointer to the supporting ablation in Appendix A.4.

---

> ### Author Response · Authors · 2026-07-06
> **Response to Reviewer 3ie3 (3/3): Training Scheme**
>
> > ### **Training Scheme**
>
> We thank the reviewer for pointing out this ambiguity, and we agree that the training scheme for each benchmark should be stated explicitly. To clarify, our experiments involve three separately trained models, and we summarize below which model is used for each benchmark.
>
> **Virtual screening (Section 4.1, MF-PCBA and ablations).** A single binary-classification model is trained on our curated PubChem BioAssay corpus (\~2.2M pairs, Section 3.3) and evaluated on MF-PCBA. All ablation variants in Section 4.1 use this same training setup.
>
> **Antibiotic discovery and prospective wet-lab validation (Sections 4.3 and 4.4).** These use the ***exact same*** PubChem-trained virtual-screening classifier described above, applied directly with **no additional training or fine-tuning** on any antibiotic data. This is precisely what we meant by "without finetuning" in Section 4.3: the model is evaluated zero-shot on a chemical and target space entirely distinct from its training distribution, which is why we regard the strong performance here (and the prospective DnaG hits) as evidence of genuine generalization rather than in-domain fitting.
>
> **Enzyme-substrate specificity (Section 4.2).** Because this task differs from binding classification, the same architecture is retrained from scratch on the ESIBank dataset, following the "unknown enzyme & substrate" split and 4-fold cross-validation protocol of EZSpecificity (Section 3.3–3.4).
>
> **Affinity regression (Appendix A).** A separate regression model (identical encoder, classification head replaced by a regression head) is trained on our curated SAIR dataset and evaluated on the OpenFE, FEP+, and CASP16 benchmarks.
>
> We apologize for the earlier lack of clarity, and we will add an explicit statement of the per-benchmark training scheme, along the lines above, to the revised manuscript.
>
> ---
>
> We hope this response could answer your questions and address your concerns, looking forward to receive your further feedback soon.

---

> > ### Comment · Reviewer_3ie3 · 2026-07-16
> > **Convincing responses**
> >
> > Thank you for your detailed responses and for thoroughly addressing all my comments. I recommend it for publication.

---

> > > ### Author Response · Authors · 2026-07-16
> > >
> > > We sincerely thank you for your positive assessment and recommendation. We greatly appreciate the time and effort you devoted to reviewing our work, and your constructive feedback has meaningfully improved the paper.

---

### Author Response · Authors · 2026-07-06
**Release of Anonymized Code, Model Weights, and Data**

We thank the reviewers for the suggestion to release our materials, and we have prepared an anonymized release to improve the reproducibility of our work during review. The full source code, including data-processing, training, and evaluation scripts, is available at https://anonymous.4open.science/r/FlashBind_preview-0032; the docking component builds on the already open-sourced FABind+ (https://github.com/QizhiPei/FABind/tree/main/FABind_plus). The trained model weights and curated dataset splits are released at https://huggingface.co/flashbind-anon/FlashBind-preview and https://huggingface.co/datasets/flashbind-anon/FlashBind-preview, respectively. We hope these resources make our pipeline fully transparent and reproducible, and upon acceptance we will release a non-anonymized version under a permissive license.

In parallel, we are currently revising the manuscript to incorporate the additional analyses and clarifications from our responses. We will upload the revised PDF as soon as it is ready and post a follow-up comment to notify the reviewers once the update is complete.

---

### Author Response · Authors · 2026-07-09
**Revised Manuscript Uploaded**

Dear Reviewers,
﻿

We sincerely thank you once again for your careful reading and constructive suggestions, which have meaningfully improved the paper. We have uploaded a revised version that incorporates the points raised during the discussion. In the main text, we now state the per-benchmark training scheme explicitly, clarify the efficiency analysis (timing boundary and library-scale cost), add a candid discussion of the affinity-regression limitation, temper our comparison with Boltz-2 to reflect the results more precisely, and expand the failure-case and PAINS-filtering details. The appendix now includes a comprehensive data-leakage analysis, a data-curation (secondary-confirmation) ablation, a per-step efficiency breakdown, run- and target-level statistical dispersion, and an explicit self-run-versus-cited provenance for every reported number, together with a Broader Impact Statement and a reproducibility commitment.
﻿

We hope these revisions adequately address your concerns, and we would be very happy to provide any further clarification.
﻿

With appreciation,

The Authors